# Learning Mixture of Gaussians with Streaming Data

**Aditi Raghunathan**
Stanford University
aditir@stanford.edu

**Prateek Jain**
Microsoft Research, India
prajain@microsoft.com

**Ravishankar Krishnaswamy**
Microsoft Research, India
rakri@microsoft.com

## Abstract

In this paper, we study the problem of learning a mixture of Gaussians with streaming data: given a stream of $N$ points in $d$ dimensions generated by an unknown mixture of $k$ spherical Gaussians, the goal is to estimate the model parameters using a single pass over the data stream. We analyze a streaming version of the popular Lloyd's heuristic and show that the algorithm estimates all the unknown centers of the component Gaussians accurately if they are sufficiently separated. Assuming each pair of centers are $C\sigma$ distant with $C = \Omega((k \log k)^{1/4}\sigma)$ and where $\sigma^2$ is the maximum variance of any Gaussian component, we show that asymptotically the algorithm estimates the centers optimally (up to certain constants); our center separation requirement matches the best known result for spherical Gaussians [18]. For finite samples, we show that a bias term based on the initial estimate decreases at $O(1/\text{poly}(N))$ rate while variance decreases at nearly optimal rate of $\sigma^2 d/N$.

Our analysis requires seeding the algorithm with a good initial estimate of the true cluster centers for which we provide an online PCA based clustering algorithm. Indeed, the asymptotic per-step time complexity of our algorithm is the optimal $d \cdot k$ while space complexity of our algorithm is $O(dk \log k)$.

In addition to the bias and variance terms which tend to 0, the hard-thresholding based updates of streaming Lloyd's algorithm is agnostic to the data distribution and hence incurs an *approximation error* that cannot be avoided. However, by using a streaming version of the classical *(soft-thresholding-based)* EM method that exploits the Gaussian distribution explicitly, we show that for a mixture of two Gaussians the true means can be estimated consistently, with estimation error decreasing at nearly optimal rate, and tending to 0 for $N \to \infty$.

## 1 Introduction

Clustering data into homogeneous clusters is a critical first step in any data analysis/exploration task and is used extensively to pre-process data, form features, remove outliers and visualize data. Due to the explosion in amount of data collected and processed, designing clustering algorithms that can handle *large datasets* that do not fit in RAM is paramount to any big-data system. A common approach in such scenarios is to treat the entire dataset as a *stream* of data, and then design algorithms which update the model after every few points from the data stream. In addition, there are several practical applications where the data itself is not available beforehand and is streaming in, for example in any typical online system like web-search.

For such a model, the algorithm of choice in practice is the so-called *streaming k-means heuristic*. It is essentially a streaming version of the celebrated $k$-means algorithm or Lloyd's heuristic [8]. The basic $k$-means algorithm is designed for offline/batch data where each data point is assigned to the nearest centroid and the centroids are then updated based on the assigned points; this process is iterated till the solution is locally optimal. The streaming version of the $k$-means algorithm assigns the new point from the stream to the closest centroid and updates this centroid *immediately*. That is, unlike offline $k$-means which first assigns all the points to the respective centroids and then updates

the centroids, the streaming algorithm updates the centroids after each point, making it much more space efficient. While streaming $k$-means and its several variants are used heavily in practice, their properties such as solution quality, time complexity of convergence have not been studied widely. In this paper, we attempt to provide such a theoretical study of the streaming $k$-means heuristic. One of the big challenges is that even the (offline) $k$-means algorithm attempts to solve a non-convex NP-hard problem. Streaming data poses additional challenges because of the large noise in each point that can deviate the solution significantly.

In the offline setting, clustering algorithms are typically studied under certain simplifying assumptions that help bypass the worst-case NP-hardness of these problems. One of the most widely studied setting is when the data is sampled from a *mixture of well-separated Gaussians* [5, 18, 1], which is also the generative assumption that we impose on the data in this work. However, the online/streaming version of the $k$-means algorithm has not been studied in such settings. In this work, we design and study a variant of the popular online $k$-means algorithm where the data is streaming-in, we cannot store more than logarithmically many data points, and each data point is sampled from a mixture of well-separated spherical Gaussians. The goal of the algorithm is then to learn the means of each of the Gaussians; note that estimating other parameters like variance, and weight of each Gaussian in the mixture becomes simple once the true means are estimated accurately.

**Our Results.** Our main contribution is the *first* bias-variance bound for the problem of learning Gaussian mixtures with streaming data. Assuming that the centers are separated by $C\sigma$ where $C = \Omega(\sqrt{\log k})$ and if we seed the algorithm with initial cluster centers that are $\leq C\sigma/20$ distance away from the true centers, then we show that the error in estimating the true centers can be decomposed into three terms and bound each one of them: **(a)** the bias term, i.e., the term dependent on distance of true means to initial centers decreases at a $1/\text{poly}(N)$ rate, where $N$ is the number of data points observed so far, **(b)** the variance term is bounded by $\sigma^2\left(\frac{d \log N}{N}\right)$ where $\sigma$ is the standard deviation of each of the Gaussians, and $d$ is the dimensionality of the data, and **(c)** an *offline approximation error*: indeed, note that even the offline Lloyd's heuristic will have an approximation error due to its hard-thresholding nature. For example, even when $k = 2$, and the centers are separated by $C\sigma$, around $\exp(-\frac{C^2}{8})$ fraction of points from the first Gaussian will be closer to the second center, and so the $k$-means heuristic will converge to centers that are at a squared distance of roughly $O(C^2)\exp(-\frac{C^2}{8})\sigma^2$ from the true means. We essentially inherit this offline error up to constants.

Note that the above result holds at a center separation of $\Omega(\sqrt{\log k}\sigma)$ distance, which is substantially weaker than the currently best-known result of $\Omega(\sigma k^{1/4})$ for even the offline problem [18]. However, as mentioned before, this only holds provided we have a good initialization. To this end, we show that when $C = \Omega(\sigma(k \log k)^{1/4})$, we can combine an *online PCA* algorithm [9, 11] with the batch $k$-means algorithm on a small seed sample of around $O(k \log k)$ points, to get such an initialization. Note that this separation requirement nearly matches the best-known result offline results [18].

Finally, we also study a soft-version of streaming $k$-means algorithm, which can also be viewed as the streaming version of the popular Expectation Maximization (EM) algorithm. We show that for mixture of two well-separated Gaussians, a variant of streaming EM algorithm recovers the above mentioned bias-variance bound but *without* the approximation error. That is, after observing infinite many samples, streaming EM converges to the true means and matches the corresponding offline results in [3, 6]; to the best of our knowledge this is also first such consistency result for the streaming mixture problem. However, the EM updates require that the data is sampled from mixture of Gaussians, while the updates of streaming Lloyd's algorithm are agnostic of the data distribution and hence same updates can be used to solve arbitrary mixture of sub-Gaussians as well.

**Technical Challenges.** One key technical challenge in analyzing streaming $k$-means algorithm in comparison to the standard streaming regression style problems is that the offline problem itself is non-convex and moreover can only be solved approximately. Hence, a careful analysis is required to separate out the error we get in each iteration in terms of the bias, variance, and inherent approximation error terms. Moreover, due to the non-convexity, we are able to guarantee decrease in error only if each of our iterates lies in a small ball around the true mean. While this is initially true due to the initialization algorithm, our intermediate centers might escape these balls during our update. However, we show using a delicate martingale based argument that with high probability, our estimates stay within *slightly larger balls* around the true means, which turns out to be sufficient for us.

**Related Work.** A closely related work to ours is an *independent* work by [17] which studies a stochastic version of $k$-means for data points that satisfy a spectral variance condition which can be seen as a deterministic version of the mixture of distributions assumption. However, their method requires multiple passes over the data, thus doesn't fit directly in the streaming $k$-means setting. In particular, the above mentioned paper analyzes the stochastic $k$-means method only for highly accurate initial set of iterates which requires a large burn-in period of $t = O(N^2)$ and hence needs $O(N)$ passes over the data, where $N$ is the number of data points. Tensor methods [1, 10] can also be extended to cluster streaming data points sampled from a mixture distribution but these methods suffer from large sample/time complexity and might not provide reasonable results when the data distribution deviates from the assumed generative model.

In addition to the gaussian mixture model, clustering problems are also studied under other models such as data with small spectral variance [12], stability of data [4], etc. It would be interesting to study the streaming versions in such models as well.

**Paper Outline.** We describe our models and problem setup in Section 2. We then present our streaming $k$-means algorithm and its proof overview in Sections 3 and 4. We then discuss the initialization procedure in Section 5. Finally we describe our streaming-EM algorithm in Section 6.

## 2 Setup and Notation

We assume that the data is drawn from a mixture of $k$ spherical Gaussians distributions, i.e.,

$$\mathbf{x^t} \overset{i.i.d}{\sim} \sum_i w_i \mathcal{N}(\boldsymbol{\mu}_i^\star, \sigma^2 I), \boldsymbol{\mu}_i^\star \in \mathbb{R}^d \ \forall i = 1, 2, \ldots k \tag{1}$$

where $\boldsymbol{\mu}_i^\star \in \mathbb{R}^d$ is the mean of the $i$-th mixture component, mixture weights $w_i \geq 0$, and $\sum_i w_i = 1$. All the problem parameters (i.e., the true means, the variance $\sigma^2$ and the mixture weights) are unknown to the algorithm. Using the standard streaming setup, where the $t^{th}$ sample $\mathbf{x^t} \in \mathbb{R}^d$ is drawn from the data distribution, our goal is to produce an estimate $\hat{\mu}_i$ of $\boldsymbol{\mu}_i^\star$ for $i = 1, 2, \ldots k$ in a single pass over the data using bounded space.

**Center Separation.** A suitable notion of signal to noise ratio for our problem turns out to be the ratio of minimum separation between the true centers and the maximum variance along any direction. We denote this ratio by $C = \min_{i,j} \frac{\|\boldsymbol{\mu}_i^\star - \boldsymbol{\mu}_j^\star\|}{\sigma}$. For convenience, we also denote $\frac{\|\boldsymbol{\mu}_i^\star - \boldsymbol{\mu}_j^\star\|}{\sigma}$ by $C_{ij}$. Here and in the rest of the paper, $\|\mathbf{y}\|$ is the Euclidean norm of a vector $\mathbf{y}$. We use $\eta$ to denote the learning rate of the streaming updates and $\boldsymbol{\mu}_i^t$ to denote the estimate of $\boldsymbol{\mu}_i^\star$ at time $t$.

**Remarks.** For a cleaner presentation, we assume that all the mixture weights are $1/k$, but our results hold with general weights as long as an appropriate center separation condition is satisfied. Secondly, our proofs also go through when the Gaussians have different variances $\sigma_i^2$, as long as the separation conditions are satisfied with $\sigma = \max_i \sigma_i$. We furnish details in the full version of this paper [14].

## 3 Algorithm and Main Result

In this section, we describe our proposed streaming clustering algorithm and present our analysis of the algorithm. At a high level, we follow the approach of various recent results for (offline) mixture recovery algorithms [18, 12]. That is, we initialize the algorithm with an SVD style operation which de-noises the data significantly in Algorithm 1 and then apply our streaming version of Lloyd's heuristic in Algorithm 2. Note that the Lloyd's algorithm is agnostic to the underlying distribution and does not include distribution specific terms like variance etc.

Intuitively, the initialization algorithm first computes an online batch PCA in the for-loop. After this step, we perform an offline distance-based clustering on the projected subspace (akin to Vempala-Wang for the offline algorithm). Note that since we only need estimates for centers within a suitable proximity from the true centers, this step only uses few (roughly $k \log k$) samples. These centers are fed as the initial centers for the streaming update algorithm. The streaming algorithm then, for each new sample, updates the current center which is closest to the sample, and iterates.

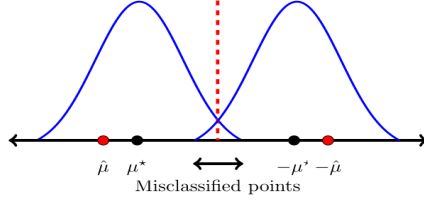

Figure 1: Illustration of optimal K-means error

| **Algorithm 1** InitAlg($N_0$) | **Algorithm 2** StreamKmeans($N, N_0$) |
|---|---|
| $U \leftarrow$ random orthonormal matrix $\in \mathbb{R}^{d \times k}$ <br> $B = \Theta(d \log d)$, $S = 0$ <br> **for** $t = 1$ to $N_0 - k \log k$ **do** <br>    **if** $\mod(t, B) = 0$ **then** <br>       $U \leftarrow QR(S \cdot U), \quad S \leftarrow 0$ <br>    **end if** <br>    Receive $\mathbf{x}^t$ as generated by the input stream <br>    $S = S + \mathbf{x}^t(\mathbf{x}^t)^T$ <br> **end for** <br> $X_0 = [\mathbf{x}^{N_0 - k \log k + 1}, \ldots, \mathbf{x}^{N_0}]$ <br> Form nearest neighbor graph using $U^T X_0$ and <br> find connected components <br> $[\nu_1^0, \ldots, \nu_k^0] \leftarrow$ mean of points in each compo-<br>nent <br> **Return:** $[\boldsymbol{\mu}_1^0, \ldots, \boldsymbol{\mu}_k^0] = [U\nu_1^0, \ldots, U\nu_k^0]$ | 1: **Set** $\eta \leftarrow \frac{3k \log 3N}{N}$. <br> 2: **Set** $\{\boldsymbol{\mu}_1^0, \ldots, \boldsymbol{\mu}_k^0\} \leftarrow$ InitAlgo($N_0$). <br> 3: **for** $t = 1$ to $N$ **do** <br> 4:   Receive $\mathbf{x}^{t+N_0}$ given by the input stream <br> 5:   $\mathbf{x} = \mathbf{x}^{t+N_0}$ <br> 6:   Let $i_t = \arg\min_i \|\mathbf{x} - \boldsymbol{\mu}_i^{t-1}\|$. <br> 7:   **Set** $\boldsymbol{\mu}_{i_t}^t = (1 - \eta)\boldsymbol{\mu}_{i_t}^{t-1} + \eta\mathbf{x}$ <br> 8:   **Set** $\boldsymbol{\mu}_i^t = \boldsymbol{\mu}_i^{t-1}$ for $i \neq i_t$ <br> 9: **end for** <br> 10: Output: $\boldsymbol{\mu}_1^N, \ldots, \boldsymbol{\mu}_k^N$ |

We now present our main result for the streaming clustering problem.

**Theorem 1.** *Let $\mathbf{x}^t$, $1 \leq t \leq N + N_0$ be generated using a mixture of Gaussians* (1) *with $w_i = 1/k$, $\forall i$. Let $N_0, N \geq O(1)k^3 d^3 \log d$ and $C \geq \Omega((k \log k)^{1/4})$. Then, the mean estimates $(\boldsymbol{\mu}_1^N, \ldots, \boldsymbol{\mu}_k^N)$ output by Algorithm 2 satisfies the following error bound:*

$$\mathbb{E}\left[\sum_i \|\boldsymbol{\mu}_i^N - \boldsymbol{\mu}_i^\star\|^2\right] \leq \underbrace{\frac{\max_i \|\boldsymbol{\mu}_i^\star\|^2}{N^{\Omega(1)}}}_{\text{bias}} + O(k^3)\left(\underbrace{\sigma^2 \frac{d \log N}{N}}_{\text{variance}} + \underbrace{\exp(-C^2/8)(C^2 + k)\sigma^2}_{\approx \text{offline } k-\text{means error}}\right).$$

Our error bound consists of three key terms: *bias, variance, and offline $k$-means error*, with bias and variance being standard statistical error terms: (i) bias is dependent on the initial estimation error and goes down at $N^\zeta$ rate where $\zeta > 1$ is a large constant; (ii) variance error is the error due to noise in each observation $\mathbf{x}^t$ and goes down at nearly optimal rate of $\approx \sigma^2 \frac{d}{N}$ albeit with an extra $\log N$ term as well as worse dependence on $k$; and (iii) an offline $k$-means error, which is the error that even the offline Lloyds' algorithm would incur for a given center separation $C$. Note that while sampling from the mixture distribution, $\approx \exp(-C^2/8)$ fraction of data-points can be closer to the true means of other clusters rather than their own mean, because the tails of the distributions overlap. Hence, in general it is not possible to assign back these points to the *correct* cluster, without any modeling assumptions. These misclassified points will shift the estimated centers along the line joining the means. See Figure 3 for an illustration. This error can however be avoided by performing soft updates, which is discussed in Section 6.

**Time, space, and sample complexity**: Our algorithm has nearly optimal time complexity of $O(d \cdot k)$ per iteration; the initialization algorithm requires about $O(d^4 k^3)$ time. Space complexity of our algorithm is $O(dk \cdot \log k)$ which is also nearly optimal. Finally, the sample complexity is $\tilde{O}(d^3 k^3)$, which is a loose upper bound and can be significantly improved by a more careful analysis. To compare, the best known sample complexity for the *offline setting* is $\tilde{O}(kd)$ [2], which is better by a factor of $(dk)^2$.

**Analysis Overview.** The proof of Theorem 1 essentially follows from the two theorems stated below: a) update analysis *given a good initialization*; b) InitAlg analysis for showing such an initialization.

**Theorem 2** (Streaming Update). *Let* $\mathbf{x^t}$, $N_0 + 1 \leq t \leq N + N_0$ *be generated using a mixture of Gaussians* (1) *with* $w_i = 1/k$, $\forall i$, *and* $N = \Omega(k^3 d^3 \log kd)$. *Also, let the center-separation* $C \geq \Omega(\sqrt{\log k})$, *and also suppose our initial centers* $\boldsymbol{\mu}_i^0$ *are such that for all* $1 \leq i \leq k$, $\|\boldsymbol{\mu}_i^0 - \boldsymbol{\mu}_i^\star\| \leq \frac{C\sigma}{20}$.

*Then, the streaming update of* StreamKmeans$(N, N_0)$ , *i.e, Steps 3-8 of Algorithm 2 satisfies:*

$$\mathbb{E}\left[\sum_i \|\boldsymbol{\mu}_i^N - \boldsymbol{\mu}_i^\star\|^2\right] \leq \frac{\max_i \|\boldsymbol{\mu}_i^\star\|^2}{N^{\Omega(1)}} + O(k^3)\left(\exp(-C^2/8)(C^2 + k)\sigma^2 + \frac{\log N}{N} d\sigma^2\right).$$

Note that our streaming update analysis requires only $C = \Omega(\sqrt{\log k})$ separation but needs appropriate initialization that is guaranteed by the below result.

**Theorem 3** (Initialization). *Let* $\mathbf{x^t}$, $1 \leq t \leq N_0$ *be generated using a mixture of Gaussians* (1) *with* $w_i = 1/k$, $\forall i$. *Let* $\boldsymbol{\mu}_1^0, \boldsymbol{\mu}_2^0, \ldots \boldsymbol{\mu}_k^0$ *be the output of Algorithm 1. If* $C = \Omega\left((k \log k)^{1/4}\right)$ *and* $N_0 = \Omega\left(d^3 k^3 \log dk\right)$, *then w.p.* $\geq 1 - 1/poly(k)$, *we have* $\max_i \|\boldsymbol{\mu}_i^0 - \boldsymbol{\mu}_i^\star\| \leq \frac{C}{20}\sigma$.

## 4 Streaming Update Analysis

At a high level our analysis shows that at each step of the streaming updates, the error decreases on average. However, due to the non-convexity of the objective function we can show such a decrease only if the current estimates of our centers lie in a small ball around the true centers of the gaussians. Indeed, while the initialization provides us with such centers, due to the added noise in each step, our iterates may occasionally fall outside these balls, and we need to bound the probability that this happens. To overcome this, we start with initial centers that are within *slightly smaller balls* around the true means, and use a careful Martingale argument to show that even if the iterates go a little farther from the true centers (due to noise), with high probability, the iterates are still within the *slightly larger ball* that we require to show decrease in error.

We therefore divide our proof in two parts: a) first we show in Section 4.1 that the error decreases in expectation, assuming that the current estimates lie in a reasonable neighborhood around the true centers; and b) in Section 4.2) we show using a martingale analysis that with high probability, each iterate satisfies the required neighborhood condition if the initialization is good enough.

We formalize the required condition for our per-iteration error analysis below. For the remainder of this section, we fix the initialization and only focus on Steps 3-8 of Algorithm 2.

**Definition 1.** *For a fixed initialization, and given a sequence of points* $\omega_t = (\mathbf{x^{t'+N_0+1}} : 0 \leq t' < t)$, *we say that condition* $\mathcal{I}_t$ *is satisfied at time* $t$ *if* $\max_i \|\boldsymbol{\mu}_i^{t'} - \boldsymbol{\mu}_i^\star\| \leq \frac{C\sigma}{10}$ *holds for all* $0 \leq t' \leq t$. *Note that given a sequence of points and a fixed initialization, Algorithm 2 is deterministic.*

We now define the following quantities which will be useful in the upcoming analysis. At any time $t \geq 1$, let $\omega_t = (\mathbf{x^{t'+N_0+1}} : 0 \leq t' < t)$ denote the sequence of points received by our algorithm. For all $t \geq 0$, let $\widetilde{E}_t^i = \|\boldsymbol{\mu}_t^i - \boldsymbol{\mu}_i^\star\|^2$ denote the random variable measuring the current error for cluster $i$, and let $\widetilde{V}_t = \max_i \widetilde{E}_t^i$ to be the maximum cluster error at time $t$. Now, let $\widehat{E}_{t+1}^i = \mathbb{E}_{\mathbf{x^{t+N_0+1}}}\left[\|\boldsymbol{\mu}_i^{t+1} - \boldsymbol{\mu}_i^\star\|^2 | \omega_t\right]$ be the *expected error* of the $i^{th}$ cluster center after receiving the $(t+1)^{th}$, conditioned on $\omega_t$. Finally, let $E_t^i = \mathbb{E}\left[\|\boldsymbol{\mu}_i^t - \boldsymbol{\mu}_i^\star\|^2 | \mathcal{I}_t\right]$ be the expected error conditioned on $\mathcal{I}_t$, and let $E_t = \sum_i E_t^i$.

### 4.1 Error Reduction in Single Iteration

Our main tool toward showing Theorem 2 is the following theorem which bounds the expected error after updating the means on arrival of the next sample.

**Theorem 4.** *If* $\mathcal{I}_t$ *holds and* $C \geq \Omega(\sqrt{\log k})$, *then for all* $i$, *we have*

$$\widehat{E}_{t+1}^i \leq (1 - \frac{\eta}{2k})\widetilde{E}_t^i + \frac{\eta}{k^5}\widetilde{V}_t + O(1)\eta^2 d\sigma^2 + O(k)\eta(1 - \eta)\exp(-C^2/8)(C^2 + k)\sigma^2.$$

*Proof sketch of Theorem 4.* In all calculations in this proof, we first assume that the candidate centers satisfy $\mathcal{I}_t$, and all expectations and probabilities are only over the new sample $\mathbf{x}^{t+N_0+1}$, which we denote by $\mathbf{x}$ after omitting the superscript. Now recall our update rule: $\boldsymbol{\mu}_i^{t+1} = (1-\eta)\boldsymbol{\mu}_i^t + \eta\mathbf{x}$ if $\boldsymbol{\mu}_i^t$ is the closest center for the new sample $\mathbf{x}$; the other centers are unchanged. To simplify notations, let:

$$g_i^t(\mathbf{x}) = 1 \text{ iff } i = \arg\min_j \|\mathbf{x} - \boldsymbol{\mu}_j^t\|, \ \ g_i^t(\mathbf{x}) = 0 \text{ otherwise.} \tag{2}$$

By definition, we have for all $i$,

$$\boldsymbol{\mu}_i^{t+1} = (1-\eta)\boldsymbol{\mu}_i^t + \eta\left(g_i^t(\mathbf{x})\mathbf{x} + (1 - g_i^t(\mathbf{x}))\boldsymbol{\mu}_i^t\right) = \boldsymbol{\mu}_i^t + \eta g_i^t(\mathbf{x})(\mathbf{x} - \boldsymbol{\mu}_i^t).$$

Our proof relies on the following simple yet crucial lemmas. The first bounds the failure probability of a sample being closest to an incorrect cluster center among our candidates. The second shows that if the candidate centers are sufficiently close to the true centers, then the failure probability of mis-classifying a point to a wrong center is (upto constant factors) the probability of mis-classification even in the optimal solution (with true centers). Finally the third lemma shows that the probability of $g_i^t(\mathbf{x}) = 1$ for each $i$ is lower-bounded. Complete details and proofs appear in [14].

**Lemma 1.** *Suppose condition $\mathcal{I}_t$ holds. For any $i$, $j \neq i$, let $\mathbf{x} \sim \text{Cl}(j)$ denote a random point from cluster $j$. Then $\Pr\left[\|\mathbf{x} - \boldsymbol{\mu}_i^t\| \leq \|\mathbf{x} - \boldsymbol{\mu}_j^t\|\right] \leq \exp(-\Omega(C_{ij}^2))$.*

**Lemma 2.** *Suppose $\max(\|\boldsymbol{\mu}_i^t - \boldsymbol{\mu}_i^\star\|, \|\boldsymbol{\mu}_i^t - \boldsymbol{\mu}_i^\star\|) \leq \sigma/C_{ij}$. For any $i$, $j \neq i$, let $x \sim \text{Cl}(j)$ denote a random point from cluster $j$. Then $\Pr\left[\|\mathbf{x} - \boldsymbol{\mu}_i^t\| \leq \|\mathbf{x} - \boldsymbol{\mu}_j^t\|\right] \leq O(1)\exp(-C_{ij}^2/8)$.*

**Lemma 3.** *If $\mathcal{I}_t$ holds and $C = \Omega(\sqrt{\log k})$, then for all $i$, then $\Pr\left[g_i^t(\mathbf{x}) = 1\right] \geq \frac{1}{2k}$.*

And so, equipped with the above notations and lemmas, we have

$$
\begin{aligned}
\widehat{E}_{t+1}^i &= \mathbb{E}_{\mathbf{x}}\left[\|\boldsymbol{\mu}_i^{t+1} - \boldsymbol{\mu}_i^\star\|^2\right] \\
&= (1-\eta)^2\|\boldsymbol{\mu}_i^t - \boldsymbol{\mu}_i^\star\|^2 + \eta^2\mathbb{E}\left[\|g_i^t(\mathbf{x})(\mathbf{x} - \boldsymbol{\mu}_i^\star) + (1 - g_i^t(\mathbf{x}))(\boldsymbol{\mu}_i^t - \boldsymbol{\mu}_i^\star)\|^2\right] \\
&\quad + 2\eta(1-\eta)\mathbb{E}\left[\left\langle\boldsymbol{\mu}_i^t - \boldsymbol{\mu}_i^\star, \left(g_i^t(\mathbf{x})(\mathbf{x} - \boldsymbol{\mu}_i^\star) + (1 - g_i^t(\mathbf{x}))(\boldsymbol{\mu}_i^t - \boldsymbol{\mu}_i^\star)\right)\right\rangle\right] \\
&\leq (1 - \tfrac{\eta}{2k})\widetilde{E}_t^i + \eta^2\underbrace{\mathbb{E}\left[\|g_i^t(\mathbf{x})(\mathbf{x} - \boldsymbol{\mu}_i^\star)\|^2\right]}_{T_1} + 2\eta(1-\eta)\underbrace{\mathbb{E}\left[\left\langle\boldsymbol{\mu}_i^t - \boldsymbol{\mu}_i^\star, \left(g_i^t(\mathbf{x})(\mathbf{x} - \boldsymbol{\mu}_i^\star)\right)\right\rangle\right]}_{T_2}
\end{aligned}
$$

The last inequality holds because of the following line of reasoning: (i) firstly, the cross term in the second squared norm evaluates to 0 due to the product $g_i^t(\mathbf{x})(1 - g_i^t(\mathbf{x}))$, (ii) $\eta^2\mathbb{E}\left[(1 - g_i^t(\mathbf{x}))\|\boldsymbol{\mu}_i^t - \boldsymbol{\mu}_i^\star\|^2\right] \leq \eta^2\widetilde{E}_t^i$, (iii) $2\eta(1-\eta)\mathbb{E}\left[\langle\boldsymbol{\mu}_i^t - \boldsymbol{\mu}_i^\star, (1 - g_i^t(\mathbf{x}))(\boldsymbol{\mu}_i^t - \boldsymbol{\mu}_i^\star)\rangle\right] \leq 2\eta(1-\eta)\widetilde{E}_t^i\Pr\left[g_i^t(\mathbf{x}) = 0\right] \leq 2\eta(1-\eta)\widetilde{E}_t^i(1 - 1/2k)$ by Lemma 3, and finally (iv) by collecting terms with coefficient $\widetilde{E}_t^i$.

The proof then roughly proceeds as follows: suppose in an ideal case, $g_i^t(\mathbf{x})$ is 1 for all points $\mathbf{x}$ generated from cluster $i$, and 0 otherwise. Then, if $\mathbf{x}$ is a random sample from cluster $i$, $T_1$ would be $d\sigma^2$, and $T_2$ would be 0. Of course, the difficulty is that $g_i^t(\mathbf{x})$ is not always as well-behaved, and so the bulk of the analysis is in carefully using Lemmas 1 and 2, and appropriately "charging" the various error terms we get to the current error $\widetilde{E}_t^i$, the variance, and the offline approximation error. $\square$

## 4.2 Ensuring Proximity Condition Via Super-Martingales

In the previous section, we saw that condition $\mathcal{I}_t = 1$ is sufficient to ensure that expected one-step error reduces at time step $t + 1$. Our next result shows that $\mathcal{I}_N = 1$ is satisfied with high probability.

**Theorem 5.** *Suppose $\max_i \|\boldsymbol{\mu}_i^0 - \boldsymbol{\mu}_i^\star\| \leq \frac{C}{20}\sigma$, then $\mathcal{I}_N = 1$ w.p $\geq 1 - (\frac{1}{\text{poly}(N)})$.*

Our argument proceeds as follows. Suppose we track the behaviour of the actual error terms $\widetilde{E}_t^i$ over time, and stop the process (call it a failure) when any of these error terms exceeds $C^2\sigma^2/100$ (recall that they are all initially smaller than $C^2\sigma^2/400$). Assuming that the process has not stopped, we show that each of these error terms has a super-martingale behaviour using Theorem 4, which

says that on average, the expected one-step error drops. Moreover, we also show that the actual one-step difference, while not bounded, has a sub-gaussian tail. Our theorem now follows by using Azuma-Hoeffding type inequality for super-martingale sequences.

## 4.3 Wrapping Up

Now, using Theorems 4 and 5, we can get the following theorem.

**Theorem 6.** *Let* $\gamma = O(k)\eta^2 d\sigma^2 + O(k^2)\eta(1-\eta)\exp(-C^2/8)(C^2 + k)\sigma^2$. *Then if* $C \geq \Omega(\sqrt{\log k})$, *for all t, we have* $E_{t+1} \leq (1 - \frac{\eta}{4k})E_t + \gamma$. *It follows that* $E_N \leq (1 - \frac{\eta}{4k})^N E_0 + \frac{4k}{\eta}\gamma$.

*Proof.* Let $\overline{E}_{t+1}^i = \mathbb{E}\left[\|\boldsymbol{\mu}_i^{t+1} - \boldsymbol{\mu}_i^\star\|^2 \,\Big|\, \mathcal{I}_t\right]$ to be the average over all sample paths of $\widetilde{E}_{t+1}^i$ conditioned on $\mathcal{I}_t$. Recall that $E_{t+1}$ is very similar, except the conditioning is on $\mathcal{I}_{t+1}$. With this notation, let us take expectation over all sample paths where $\mathcal{I}_t$ is satisfied, and use Theorem 4 to get

$$\overline{E}_{t+1}^i \leq (1 - \frac{\eta}{2k})E_t^i + \frac{\eta}{k^5}E_t + O(1)\eta^2 d\sigma^2 + O(k)\eta(1-\eta)\exp(-C^2/8)(C^2 + k)\sigma^2\,.$$

And so, summing over all $i$ we will get

$$\overline{E}_{t+1} \leq (1 - \frac{\eta}{3k})E_t + O(k)\eta^2 d\sigma^2 + O(k^2)\eta(1-\eta)\exp(-C^2/8)(C^2 + k)\sigma^2\,.$$

Finally note that $E_{t+1}$ and $\overline{E}_{t+1}$ are related as $E_{t+1}\Pr[\mathcal{I}_{t+1}] \leq \overline{E}_{t+1}\Pr[\mathcal{I}_t]$, and so $E_{t+1} \leq \overline{E}_{t+1}(1 + \frac{1}{N^2})$ since $\Pr[\mathcal{I}_{t+1}] \geq 1 - 1/N^5$ by Theorem 5. $\qquad\square$

*Proof of Theorem 2.* From Theorem 5 we know that the probability of $\mathcal{I}_N$ being satisfied is $1 - 1/N^5$, and in this case, we can use Theorem 6 to get the desired error bound. In case $\mathcal{I}_N$ fails, then the maximum possible error is roughly $\max_{i,j}\|\boldsymbol{\mu}_i^\star - \boldsymbol{\mu}_j^\star\|^2 \cdot N$ (when all our samples are sent to the same cluster), which contributes a negligible amount to the bias term. $\qquad\square$

## 5 Initialization for streaming k-means

In Section 4 we saw that our proposed streaming algorithm can lead to a good solution for any separation $C\sigma \geq O(\sqrt{\log k})\sigma$ if we can initialize all centers such that $\|\boldsymbol{\mu}_i^0 - \boldsymbol{\mu}_i^\star\| \leq \frac{C}{20}\sigma$. We now show that InitAlg (Algorithm 1) is one such procedure. We first approximately compute the top-$k$ eigenvectors $U$ of the data covariance *using a streaming PCA* algorithm [9, 13] on $O(k^3 d^3 \log d)$ samples. We next store $k \log k$ points and project them onto the subspace spanned by $U$. We then perform a simple distance based clustering [18] that correctly clusters the stored points (assuming reasonable center separation), and finally we output these cluster centers.

*Proof of Theorem 3.* Using an argument similar to [9] (Theorem 3), we get that $U$ obtained by the online PCA algorithm (Steps 1:4 of Algorithm 1) satisfies (w.p. $\geq 1 - 1/poly(d)$):

$$\|UU^T\boldsymbol{\mu}_i^\star - \boldsymbol{\mu}_i^\star\|^2 \leq .01\sigma^2, \ \forall 1 \leq i \leq k. \tag{3}$$

Now, let $\widehat{\boldsymbol{\mu}}_i^* = U^T\boldsymbol{\mu}_i^\star$. For any $\mathbf{x}$ sampled from mixture distribution (1), $U^T\mathbf{x} \sim \sum_i w_i \mathcal{N}(\widehat{\boldsymbol{\mu}}_i^*, \sigma^2 I)$. Hence, if $U^T\mathbf{x}^t, U^T\mathbf{x}^{t'}$ both belong to cluster $i$, then (w.p. $\geq 1 - 1/k^\alpha$):

$$\|U^T\mathbf{x}^t - U^T\mathbf{x}^{t'}\|^2 = \|U^T(\mathbf{z}^t - \mathbf{z}^{t'})\|_2^2 \leq (k + 8\alpha\sqrt{k \log k})\sigma^2, \tag{4}$$

where $\mathbf{x}^t = \boldsymbol{\mu}_i^\star + \mathbf{z}^t$ and $\mathbf{x}^{t'} = \boldsymbol{\mu}_i^\star + \mathbf{z}^{t'}$. The last inequality above follows by using standard $\chi^2$ random variable tail bound. Similarly if $U^T\mathbf{x}^t, U^T\mathbf{x}^{t'}$ belong to cluster $i$ and $j$, i.e., $\mathbf{x}^t = \boldsymbol{\mu}_i^\star + \mathbf{z}^t$ and $\mathbf{x}^{t'} = \boldsymbol{\mu}_j^\star + \mathbf{z}^{t'}$ then (w.p. $\geq 1 - 1/k^\alpha$):

$$\|U^T\mathbf{x}^t - U^T\mathbf{x}^{t'}\|^2 = \|\widehat{\boldsymbol{\mu}}_i^* - \widehat{\boldsymbol{\mu}}_j^*\|^2 + \|U^T(\mathbf{z}^t - \mathbf{z}^{t'})\|_2^2 + 2(\widehat{\boldsymbol{\mu}}_i^* - \widehat{\boldsymbol{\mu}}_j^*)^T U^T(\mathbf{z}^t - \mathbf{z}^{t'})$$
$$\geq (C^2 - .2C + 8\alpha\sqrt{k \log k} - 16\alpha C\sqrt{\log k})\sigma^2, \tag{5}$$

where the above equation follows by using (3), setting $\alpha = C/32$ and using $C = \Omega((k \log k)^{1/4})$.

Using (4), (5), w.h.p. all the points from the same cluster are closer to each other than points from other clusters. Hence, connected components of nearest neighbor graph recover clusters accurately.

Now, we estimate $\widehat{\boldsymbol{\mu}}_i = \frac{1}{|Cluster(i)|} \sum_{t \in Cluster(i)} U^T \mathbf{x}^t$ for each $i$. Since, our clustering is completely accurate, we have w.p. $\geq 1 - 2m^2/k^{C/32}$,

$$\|\widehat{\boldsymbol{\mu}}_i - \widehat{\boldsymbol{\mu}}_i^*\|_2 \leq \sigma \frac{\sqrt{\log k}}{\sqrt{|Cluster(i)|}}. \tag{6}$$

As $w_i = 1/k$ for all $i$, $|Cluster(i)| \geq \frac{m}{k} - C\sqrt{\frac{m}{k}}$ w.p. $\geq 1 - 1/k^{C/32}$. Theorem now follows by setting $m = O(k \log k)$ and by using (3), (6) along with $C = \Omega((k \log k)^{1/4})$. $\qquad\square$

**Remark 1.** *We would like to emphasize that our analysis for the convergence of streaming algorithms works even for smaller separations $C = O(\sqrt{\log k})$, as long as we can get a good enough initialization. Hence, a better initialization algorithm with weaker dependence of $C$ on $k$ would lead to an improvement in the overall algorithm.*

## 6  Soft thresholding EM based algorithm

In this section, we study a streaming version of the Expectation Maximization (EM) algorithm [7] which is also used extensively in practice. While the standard $k$-means or Lloyd's heuristic is known to be agnostic to the distribution, and the same procedure can solve the mixture problem for a variety of distributions [12], EM algorithms are designed specifically for the input mixture distribution. In this section, we consider a streaming version of the EM algorithm when applied to the problem of mixture of two spherical Gaussians with known variances. In this case, the EM algorithm reduces to a softer version of the Lloyd's algorithm where a point can be *partially* assigned to the two clusters. Recent results by [6, 3, 19] show convergence of the EM algorithm in the offline setting for this simple setup. In keeping with earlier notation, let $\boldsymbol{\mu}_1^\star = \boldsymbol{\mu}^\star$ and $\boldsymbol{\mu}_2^\star = -\boldsymbol{\mu}^\star$ and the center separation $C = \frac{2\|\boldsymbol{\mu}^\star\|}{\sigma}$. Hence, $\mathbf{x^t} \overset{i.i.d}{\sim} \frac{1}{2}\mathcal{N}(\boldsymbol{\mu}^\star, \sigma^2 I) + \frac{1}{2}\mathcal{N}(-\boldsymbol{\mu}^\star, \sigma^2 I)$.

---

**Algorithm 3** StreamSoftUpdate$(N, N_0)$

---

> **Set** $\eta = \frac{3 \log N}{N}$.
> **Set** $\boldsymbol{\mu}_i^0 \leftarrow$ InitAlgo$(N_0)$.
> **for** $t = 1$ to $N$ **do**
>    Receive $\mathbf{x}^{t+N_0}$ as generated by the input stream.
>    $\mathbf{x} = \mathbf{x}^{t+N_0}$
>    **Let** $w_t = \frac{\exp\left(\frac{-\|\mathbf{x} - \boldsymbol{\mu}^t\|^2}{\sigma^2}\right)}{\exp\left(\frac{-\|\mathbf{x} - \boldsymbol{\mu}^t\|^2}{\sigma^2}\right) + \exp\left(\frac{-\|\mathbf{x} + \boldsymbol{\mu}^t\|^2}{\sigma^2}\right)}$
>
>    **Set** $\boldsymbol{\mu}^{t+1} = (1 - \eta)\boldsymbol{\mu}^t + \eta[2w_t - 1]\mathbf{x}$.
> **end for**

---

In our algorithm, $w_t(\mathbf{x})$ is an estimate of the probability that $\mathbf{x}$ belongs to the cluster with $\boldsymbol{\mu}^t$, given that it is drawn from a balanced mixture of gaussians at $\boldsymbol{\mu}^t$ and $-\boldsymbol{\mu}^t$. Calculating $w_t(\mathbf{x})$ corresponds to the E step and updating the estimate of the centers corresponds to the M step of the EM algorithm. Similar to the streaming Lloyd's algorithm presented in Section 3, our analysis of streaming soft updates can be separated into streaming update analysis and analysis InitAlg (which is already presented in Section 5). We now provide our main theorem, and the proof is presented in Appendix C.

**Theorem 7** (Streaming Update). *Let $\mathbf{x}^t$, $1 \leq t \leq N + N_0$ be generated using a mixture two balanced spherical Gaussians with variance $\sigma^2$. Also, let the center-separation $C \geq 4$, and also suppose our initial estimate $\boldsymbol{\mu}^0$ is such that $\|\boldsymbol{\mu}^0 - \boldsymbol{\mu}^\star\| \leq \frac{C\sigma}{20}$. Then, the streaming update of* StreamSoftUpdate$(N, N_0)$ *, i.e, Steps 3-8 of Algorithm 3 satisfies:*

$$\mathbb{E}\left[\|\boldsymbol{\mu}^N - \boldsymbol{\mu}^\star\|^2\right] \leq \underbrace{\frac{\|\boldsymbol{\mu}^\star\|^2}{N^{\Omega(1)}}}_{\text{bias}} + \underbrace{O(1)\frac{\log N}{N}d\sigma^2}_{\text{variance}}.$$

**Remark 2.** *Our bias and variance terms are similar to the ones in Theorem 1 but the above bound does not have the additional approximation error term. Hence, in this case we can estimate $\boldsymbol{\mu}^\star$ consistently but the algorithm applies only to a mixture of Gaussians while our algorithm and result in Section 3 can potentially be applied to arbitrary sub-Gaussian distributions.*

**Remark 3.** *We note that for our streaming soft update algorithm, it is not critical to know the variance $\sigma^2$ beforehand. One could get a good estimate of $\sigma$ by taking the mean of a random projection of a small number of points. We provide the details in the full version of this paper [14].*

## 7   Conclusions

In this paper, we studied the problem of clustering with streaming data where each data point is sampled from a mixture of spherical Gaussians. For this problem, we study two algorithms that use appropriate initialization: a) a streaming version of Lloyd's method, b) a streaming EM method. For both the methods we show that we can accurately initialize the cluster centers using an online PCA based method. We then show that assuming $\Omega((k \log k)^{1/4}\sigma)$ separation between the cluster centers, the updates by both the methods lead to decrease in both the bias as well as the variance error terms. For Lloyd's method there is an additional estimation error term, which even the offline algorithm incurs, and which is avoided by the EM method. However, the streaming Lloyd's method is agnostic to the data distribution and can in fact be applied to any mixture of sub-Gaussians problem. For future work, it would be interesting to study the streaming data clustering problem under deterministic assumptions like [12, 16]. Also, it is an important question to understand the optimal separation assumptions needed for even the offline gaussian mixture clustering problem.

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
