[Supplementary Material]

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

| $B = \Theta(d \log d), S = 0$ | 2: **Set** $\{\boldsymbol{\mu}_1^0, \dots, \boldsymbol{\mu}_k^0\} \leftarrow$ InitAlgo($N_0$). |
| **for** $t = 1$ to $N_0 - k \log k$ **do** | 3: **for** $t = 1$ to $N$ **do** |
| $\quad$ **if** $\mod(t, B) = 0$ **then** | 4: $\quad$ Receive $\mathbf{x}^{t+N_0}$ given by the input stream |
| $\quad\quad U \leftarrow QR(S \cdot U), \quad S \leftarrow 0$ | 5: $\quad \mathbf{x} = \mathbf{x}^{t+N_0}$ |
| $\quad$ **end if** | 6: $\quad$ Let $i_t = \arg\min_i \|\mathbf{x} - \boldsymbol{\mu}_i^{t-1}\|$. |
| $\quad$ Receive $\mathbf{x}^t$ as generated by the input stream | 7: $\quad$ **Set** $\boldsymbol{\mu}_{i_t}^t = (1 - \eta)\boldsymbol{\mu}_{i_t}^{t-1} + \eta\mathbf{x}$ |
| $\quad S = S + \mathbf{x}^t(\mathbf{x}^t)^T$ | 8: $\quad$ **Set** $\boldsymbol{\mu}_i^t = \boldsymbol{\mu}_i^{t-1}$ for $i \neq i_t$ |
| **end for** | 9: **end for** |
| $X_0 = [\mathbf{x}^{N_0 - k \log k + 1}, \dots, \mathbf{x}^{N_0}]$ | 10: Output: $\boldsymbol{\mu}_1^N, \dots, \boldsymbol{\mu}_k^N$ |

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

# A  Proofs from Section 4

*Proof of Lemma 1.* Let $\mathbf{x} = \boldsymbol{\mu}_j^\star + \mathbf{z}$ where $\mathbf{z}$ is a mean-$\mathbf{0}$ spherical Gaussian with variance $\sigma^2 I$. Now, the condition $\|\mathbf{x} - \boldsymbol{\mu}_i^t\| < \|\mathbf{x} - \boldsymbol{\mu}_j^t\|$ is equivalent to:

$$\|\boldsymbol{\mu}_j^\star + \mathbf{z} - \boldsymbol{\mu}_i^t\| < \|\boldsymbol{\mu}_j^\star + \mathbf{z} - \boldsymbol{\mu}_j^t\|, i.e., \|\boldsymbol{\mu}_j^\star - \boldsymbol{\mu}_i^\star + \mathbf{z} + \boldsymbol{\mu}_i^\star - \boldsymbol{\mu}_i^t\|^2 < \|\boldsymbol{\mu}_j^\star + \mathbf{z} - \boldsymbol{\mu}_j^t\|^2.$$

Now expanding the squared-norm, we get this to be equivalent to:

$$\|\boldsymbol{\mu}_j^\star - \boldsymbol{\mu}_i^\star\|^2 + \|\boldsymbol{\mu}_i^\star - \boldsymbol{\mu}_i^t\|^2 + 2\langle \mathbf{z}, (\boldsymbol{\mu}_j^\star - \boldsymbol{\mu}_i^\star) + (\boldsymbol{\mu}_i^\star - \boldsymbol{\mu}_i^t) - (\boldsymbol{\mu}_j^\star - \boldsymbol{\mu}_j^t)\rangle + 2\langle\boldsymbol{\mu}_j^\star - \boldsymbol{\mu}_i^\star, \boldsymbol{\mu}_i^\star - \boldsymbol{\mu}_i^t\rangle < \|\boldsymbol{\mu}_j^\star - \boldsymbol{\mu}_j^t\|^2.$$

Re-arranging terms, we get:

$$\|\mathbf{x} - \boldsymbol{\mu}_i^t\| < \|\mathbf{x} - \boldsymbol{\mu}_j^t\| \text{ iff } 2 \cdot \langle \mathbf{z}, (\boldsymbol{\mu}_j^\star - \boldsymbol{\mu}_i^\star) + (\boldsymbol{\mu}_i^\star - \boldsymbol{\mu}_i^t) - (\boldsymbol{\mu}_j^\star - \boldsymbol{\mu}_j^t)\rangle$$
$$< -\|\boldsymbol{\mu}_j^\star - \boldsymbol{\mu}_i^\star\|^2 - \|\boldsymbol{\mu}_i^\star - \boldsymbol{\mu}_i^t\|^2 - 2\langle\boldsymbol{\mu}_j^\star - \boldsymbol{\mu}_i^\star, \boldsymbol{\mu}_i^\star - \boldsymbol{\mu}_i^t\rangle + \|\boldsymbol{\mu}_j^\star - \boldsymbol{\mu}_j^t\|^2. \quad (7)$$

Using the Cauchy-Schwarz, we know that the above holds if the following is true:

$$2\langle\mathbf{z}, (\boldsymbol{\mu}_j^\star - \boldsymbol{\mu}_i^\star) + (\boldsymbol{\mu}_i^\star - \boldsymbol{\mu}_i^t) - (\boldsymbol{\mu}_j^\star - \boldsymbol{\mu}_j^t)\rangle < -\|\boldsymbol{\mu}_j^\star - \boldsymbol{\mu}_i^\star\|^2 - \|\boldsymbol{\mu}_i^\star - \boldsymbol{\mu}_i^t\|^2 + 2\|\boldsymbol{\mu}_j^\star - \boldsymbol{\mu}_i^\star\|\|\boldsymbol{\mu}_i^\star - \boldsymbol{\mu}_i^t\|$$
$$+ \|\boldsymbol{\mu}_j^\star - \boldsymbol{\mu}_j^t\|^2.$$

Using $\mathcal{I}_t$ and the fact that $\|\boldsymbol{\mu}_j^\star - \boldsymbol{\mu}_i^\star\|^2 = C_{ij}^2\sigma^2$, the RHS above is at most $-\frac{3}{4}C_{ij}^2\sigma^2$. Moreover, the LHS is a Gaussian with mean 0, and variance at most $4\sigma^2\|(\boldsymbol{\mu}_j^\star - \boldsymbol{\mu}_i^\star) + (\boldsymbol{\mu}_i^\star - \boldsymbol{\mu}_i^t) - (\boldsymbol{\mu}_j^\star - \boldsymbol{\mu}_j^t)\|^2 \leq \sigma^2(C_{ij}\sigma + 2(C\sigma/10))^2$. So applying standard Gaussian concentration yields the desired result. $\square$

*Proof of Lemma 2.* Following the proof of Lemma 1, we get that using Cauchy-Schwarz inequality, the desired condition is stronger than the following condition:

$$2\langle\mathbf{z}, (\boldsymbol{\mu}_j^\star - \boldsymbol{\mu}_i^\star) + (\boldsymbol{\mu}_i^\star - \boldsymbol{\mu}_i^t) - (\boldsymbol{\mu}_j^\star - \boldsymbol{\mu}_j^t)\rangle < -\|\boldsymbol{\mu}_j^\star - \boldsymbol{\mu}_i^\star\|^2 - \|\boldsymbol{\mu}_i^\star - \boldsymbol{\mu}_i^t\|^2$$
$$+ 2\|\boldsymbol{\mu}_j^\star - \boldsymbol{\mu}_i^\star\|\|\boldsymbol{\mu}_i^\star - \boldsymbol{\mu}_i^t\| + \|\boldsymbol{\mu}_j^\star - \boldsymbol{\mu}_j^t\|^2.$$

But now, since we have much better bounds on $\|\boldsymbol{\mu}_i^t - \boldsymbol{\mu}_i^\star\| < \sigma/C_{ij}$ and $\|\boldsymbol{\mu}_j^t - \boldsymbol{\mu}_j^\star\| < \sigma/C_{ij}$, we will get that the probability is almost equivalent to that of $\mathcal{N}(0, 4C_{ij}^2\sigma^4)$ random variable is smaller than $-C_{ij}^2\sigma^2$, which in turn is $O(1)\exp(-C_{ij}^2/8)$. $\square$

*Proof of Lemma 3.* The proof follows directly from Lemma 1. Consider the case when the next point $\mathbf{x}$ is sampled from cluster $i$. In this case, for every $j \neq i$, the probability that $g_j^t(\mathbf{x}) = 1$ is at most $1/2k$ by Lemma 1 since $C \geq \Omega(\sqrt{\log k})$. Then by the union bound, we get that that with probability at least $1/2$, $g_i^t(\mathbf{x}) = 1$. Now since the point $\mathbf{x}$ is sampled from cluster $i$ with probability $1/k$, the proof follows. $\square$

*Proof of Theorem 4.* In all calculations in this proof, we first assume that the candidate centers satisfy $\mathcal{I}_t$, and all expectations and probabilities are only over the new sample $\mathbf{x}^t$. For brevity in notation, we omit the $t$ superscript and simply refer to the random sample by $\mathbf{x}$.

$$\widehat{E}_{t+1}^i = \mathbb{E}_{\mathbf{x}}\left[\|\boldsymbol{\mu}_i^{t+1} - \boldsymbol{\mu}_i^\star\|^2\right]$$
$$= (1-\eta)^2\|\boldsymbol{\mu}_i^t - \boldsymbol{\mu}_i^\star\|^2 + \eta^2\mathbb{E}\left[\|g_i^t(\mathbf{x})(\mathbf{x} - \boldsymbol{\mu}_i^\star) + (1 - g_i^t(\mathbf{x}))(\boldsymbol{\mu}_i^t - \boldsymbol{\mu}_i^\star)\|^2\right]$$
$$+ 2\eta(1-\eta)\mathbb{E}\left[\left\langle\boldsymbol{\mu}_i^t - \boldsymbol{\mu}_i^\star, \left(g_i^t(\mathbf{x})(\mathbf{x} - \boldsymbol{\mu}_i^\star) + (1 - g_i^t(\mathbf{x}))(\boldsymbol{\mu}_i^t - \boldsymbol{\mu}_i^\star)\right)\right\rangle\right]$$
$$\leq (1 - \frac{\eta}{2k})\widetilde{E}_t^i + \eta^2 \underbrace{\mathbb{E}\left[\|g_i^t(\mathbf{x})(\mathbf{x} - \boldsymbol{\mu}_i^\star)\|^2\right]}_{T_1} + 2\eta(1-\eta)\underbrace{\mathbb{E}\left[\left\langle\boldsymbol{\mu}_i^t - \boldsymbol{\mu}_i^\star, \left(g_i^t(\mathbf{x})(\mathbf{x} - \boldsymbol{\mu}_i^\star)\right)\right\rangle\right]}_{T_2}$$

The last inequality holds because of the following line of reasoning: (i) firstly, the cross term in the second squared norm evaluates to 0 due to the product $g_i^t(\mathbf{x})(1 - g_i^t(\mathbf{x}))$, (ii) $\eta^2\mathbb{E}\left[(1 - g_i^t(\mathbf{x}))\|\boldsymbol{\mu}_i^t - \boldsymbol{\mu}_i^\star\|^2\right] \leq \eta^2\widetilde{E}_t^i$, (iii) $2\eta(1-\eta)\mathbb{E}\left[\langle\boldsymbol{\mu}_i^t - \boldsymbol{\mu}_i^\star, (1 - g_i^t(\mathbf{x}))(\boldsymbol{\mu}_i^t - \boldsymbol{\mu}_i^\star)\rangle\right] \leq 2\eta(1-\eta)\widetilde{E}_t^i \Pr\left[g_i^t(\mathbf{x}) = 0\right] \leq 2\eta(1-\eta)\widetilde{E}_t^i(1 - 1/2k)$ by Lemma 3, and finally (iv) by collecting terms with coefficient $\widetilde{E}_t^i$. The proof then follows from the below two lemmas.

**Lemma 4.** *If $\mathcal{I}_t$ holds and $C = \Omega(\sqrt{\log k})$, then $T_1 \leq O(d)\sigma^2$.*

**Lemma 5.** *If $\mathcal{I}_t$ holds and $C = \Omega(\sqrt{\log k})$, then $T_2 \leq \frac{\widetilde{V}_t}{k^5} + O(k)\exp(-C^2/8)(C^2 + k)\sigma^2$.*

$\square$

*Proof of Lemma 4.*

$$
\begin{aligned}
T_1 =& \frac{1}{k}\sum_{j \neq i} \Pr\left[g_i^t(\mathbf{x}) = 1 \,\middle|\, x \sim \mathrm{Cl}(j)\right] \mathbb{E}\left[\|\mathbf{x} - \boldsymbol{\mu}_i^\star\|^2 \,\middle|\, g_t^t(\mathbf{x}) = 1 \text{ and } x \sim \mathrm{Cl}(j)\right], \\
&+ \frac{1}{k}\Pr\left[g_i^t(\mathbf{x}) = 1 \,\middle|\, x \sim \mathrm{Cl}(i)\right] \mathbb{E}\left[\|\mathbf{x} - \boldsymbol{\mu}_i^\star\|^2 \,\middle|\, g_t^t(\mathbf{x}) = 1 \text{ and } x \sim \mathrm{Cl}(i)\right], \\
\leq& \frac{1}{k}\sum_{j \neq i} O(1)\exp(-\Omega(C^2))(C^2 + d)\sigma^2 \\
&\qquad + \frac{1}{k}\Pr\left[g_i^t(\mathbf{x}) = 1 \,\middle|\, x \sim \mathrm{Cl}(i)\right] \mathbb{E}\left[\|\mathbf{x} - \boldsymbol{\mu}_i^\star\|^2 \,\middle|\, g_t^t(\mathbf{x}) = 1 \text{ and } x \sim \mathrm{Cl}(i)\right], \\
\leq& \, O(1)\exp(-\Omega(C^2))(C^2 + d)\sigma^2 + d\sigma^2.
\end{aligned}
$$

Above, the first inequality follows from Lemma 6 stated below, and the second inequality follows because we are summing a non-negative quantity over all $\mathbf{x} \sim \mathrm{Cl}(i)$ instead of only those where $g_i^t(\mathbf{x}) = 1$. $\square$

*Proof of Lemma 5.*

$$
\begin{aligned}
T_2 =& \frac{1}{k}\sum_{j \neq i} \Pr\left[g_i^t(\mathbf{x}) = 1 \,\middle|\, x \sim \mathrm{Cl}(j)\right] \mathbb{E}\left[\left\langle \boldsymbol{\mu}_i^t - \boldsymbol{\mu}_i^\star, (\mathbf{x} - \boldsymbol{\mu}_i^\star)\right\rangle \,\middle|\, g_t^t(\mathbf{x}) = 1 \text{ and } x \sim \mathrm{Cl}(j)\right] \\
&+ \frac{1}{k}\Pr\left[g_i^t(\mathbf{x}) = 1 \,\middle|\, x \sim \mathrm{Cl}(i)\right] \mathbb{E}\left[\left\langle \boldsymbol{\mu}_i^t - \boldsymbol{\mu}_i^\star, (\mathbf{x} - \boldsymbol{\mu}_i^\star)\right\rangle \,\middle|\, g_t^t(\mathbf{x}) = 1 \text{ and } x \sim \mathrm{Cl}(i)\right], \\
=& \frac{1}{k}\sum_{j \neq i} \Pr\left[g_i^t(\mathbf{x}) = 1 \,\middle|\, x \sim \mathrm{Cl}(j)\right] \mathbb{E}\left[\left\langle \boldsymbol{\mu}_i^t - \boldsymbol{\mu}_i^\star, (\mathbf{x} - \boldsymbol{\mu}_i^\star)\right\rangle \,\middle|\, g_t^t(\mathbf{x}) = 1 \text{ and } x \sim \mathrm{Cl}(j)\right] \\
&- \frac{1}{k}\Pr\left[g_i^t(\mathbf{x}) = 0 \,\middle|\, x \sim \mathrm{Cl}(i)\right] \mathbb{E}\left[\left\langle \boldsymbol{\mu}_i^t - \boldsymbol{\mu}_i^\star, (\mathbf{x} - \boldsymbol{\mu}_i^\star)\right\rangle \,\middle|\, g_t^t(\mathbf{x}) = 0 \text{ and } x \sim \mathrm{Cl}(i)\right], \\
\leq& \frac{\widetilde{V}_t}{k^5} + O(k)\exp(-C^2/8)(C^2 + k)\sigma^2.
\end{aligned}
$$

The inequality above follows from Lemmas 7, and 8. $\square$

**Lemma 6.** *Suppose $\mathcal{I}_t$ holds, and fix any $i, j \neq i$. Then, if $\mathbf{x} \sim \mathrm{Cl}(j)$, we have $\Pr_{\mathbf{x}}\left[g_i^t(\mathbf{x}) = 1\right] \mathbb{E}\left[\|\mathbf{x} - \boldsymbol{\mu}_i^\star\|^2 \,\middle|\, g_t^t(\mathbf{x}) = 1\right] \leq O(1)\exp(-\Omega(C^2))(C^2 + d)\sigma^2.$*

*Proof.* Intuitively, we know from Lemma 1 that if $\mathbf{x} \sim \mathrm{Cl}(j)$, then $\Pr_{\mathbf{x}}\left[g_i^t(\mathbf{x}) = 1\right] \leq \Pr_{\mathbf{x}}\left[\|\mathbf{x} - \boldsymbol{\mu}_i^t\| < \|\mathbf{x} - \boldsymbol{\mu}_j^t\|\right]$, which in turn is at most $\exp(-\Omega(C_{ij}^2))$. In this case, we incur a cost of $\|\mathbf{x} - \boldsymbol{\mu}_i^\star\|^2$ which is roughly $O(1)(\|\boldsymbol{\mu}_j^\star - \boldsymbol{\mu}_i^\star\|^2 + \|\mathbf{x} - \boldsymbol{\mu}_j^\star\|^2)$. The first term is $C_{ij}^2\sigma^2$, and the second term in expectation is $d\sigma^2$. Multiplying this with the probability would establish the result. Of course, the expectation calculated here is not precise due to the conditioning involved. We formally now formally show the details.

As $\|\mathbf{x} - \boldsymbol{\mu}_i^\star\|^2$ is a non-negative quantity and the condition $\|\mathbf{x} - \boldsymbol{\mu}_i^t\| < \|\mathbf{x} - \boldsymbol{\mu}_j^t\|$ is weaker than $g_i^t(\mathbf{x}) = 1$, we have: We have:

$$
\begin{aligned}
\Pr_{\mathbf{x}}\left[g_i^t(\mathbf{x}) = 1\right] \cdot \mathbb{E}\left[\|\mathbf{x} - \boldsymbol{\mu}_i^\star\|^2 \,\middle|\, g_t^t(\mathbf{x}) = 1\right] \leq& \\
\Pr_{\mathbf{x}}\left[\|\mathbf{x} - \boldsymbol{\mu}_i^t\| < \|\mathbf{x} - \boldsymbol{\mu}_j^t\|\right] \cdot& \mathbb{E}\left[\|\mathbf{x} - \boldsymbol{\mu}_i^\star\|^2 \,\middle|\, \|\mathbf{x} - \boldsymbol{\mu}_i^t\| < \|\mathbf{x} - \boldsymbol{\mu}_j^t\|\right].
\end{aligned}
$$

So we bound the RHS above to complete the proof.

Now, let $\mathbf{x} = \boldsymbol{\mu}_j^\star + \mathbf{z}$ where $\mathbf{z}$ is sampled from a spherical normal Gaussian with mean $0$ and variance $\sigma^2$ along each direction. Then note that:

$$\|\mathbf{x} - \boldsymbol{\mu}_i^\star\|^2 = \|\mathbf{z} + \boldsymbol{\mu}_j^\star - \boldsymbol{\mu}_i^\star\|^2 \le 2\|\mathbf{z}\|^2 + 2C_{ij}^2\sigma^2.$$

Moreover, we know by Lemma 1 that $\Pr_{\mathbf{x}}\left[\|\mathbf{x} - \boldsymbol{\mu}_i^t\| < \|\mathbf{x} - \boldsymbol{\mu}_j^t\|\right] \le \exp(-\Omega(C_{ij}^2))$. So multiplying the conditional expectation with the probability gives:

$$\Pr_{\mathbf{x}}\left[\|\mathbf{x} - \boldsymbol{\mu}_i^t\| < \|\mathbf{x} - \boldsymbol{\mu}_j^t\|\right] \cdot \mathbb{E}\left[2C_{ij}^2\sigma^2 \,\Big|\, \|\mathbf{x} - \boldsymbol{\mu}_i^t\| < \|\mathbf{x} - \boldsymbol{\mu}_j^t\|\right]$$
$$\le 2C_{ij}^2\sigma^2 \exp(-\Omega(C_{ij}^2)) \le 2C^2\sigma^2 \exp(-\Omega(C^2)). \quad (8)$$

We now bound: $2\Pr_{\mathbf{x}}\left[\|\mathbf{x} - \boldsymbol{\mu}_i^t\| < \|\mathbf{x} - \boldsymbol{\mu}_j^t\|\right]\mathbb{E}\left[\|\mathbf{z}\|^2 \,\Big|\, \|\mathbf{x} - \boldsymbol{\mu}_i^t\| < \|\mathbf{x} - \boldsymbol{\mu}_j^t\|\right]$. The crux of the proof now lies in the fact that the condition $\|\mathbf{x} - \boldsymbol{\mu}_i^t\|^2 < \|\mathbf{x} - \boldsymbol{\mu}_j^t\|^2$ boils down to the following linear inequality on $\mathbf{z}$:

$$2\langle \mathbf{z}, \boldsymbol{\mu}_j^t - \boldsymbol{\mu}_i^t\rangle < \|\boldsymbol{\mu}_j^t\|^2 - \|\boldsymbol{\mu}_i^t\|^2 - 2\langle \boldsymbol{\mu}_j^\star, \boldsymbol{\mu}_j^t - \boldsymbol{\mu}_i^t\rangle.$$

Inspired by this, let us define $\tau := \frac{1}{2\|\boldsymbol{\mu}_j^t - \boldsymbol{\mu}_i^t\|}\left(\|\boldsymbol{\mu}_j^t\|^2 - \|\boldsymbol{\mu}_i^t\|^2 - 2\langle \boldsymbol{\mu}_j^\star, \boldsymbol{\mu}_j^t - \boldsymbol{\mu}_i^t\rangle\right)$, and also define $\mathbf{a} := \frac{\boldsymbol{\mu}_j^t - \boldsymbol{\mu}_i^t}{\|\boldsymbol{\mu}_j^t - \boldsymbol{\mu}_i^t\|}$. So now note that the condition $\|\mathbf{x} - \boldsymbol{\mu}_i^t\|^2 < \|\mathbf{x} - \boldsymbol{\mu}_j^t\|^2$ is equivalent to $\langle \mathbf{z}, \mathbf{a}\rangle < \tau$.

Since $\mathbf{z}$ is a spherical Gaussian, imposing a condition on $\langle \mathbf{z}, \mathbf{a}\rangle < \tau$ results in a truncated Gaussian along the direction $\mathbf{a}$ and an independent Gaussian in all orthogonal directions. So the expected squared length of the projection of $\mathbf{z}$ along orthogonal directions is $\sigma^2$. Since we know that $\Pr_{\mathbf{x}}\left[\|\mathbf{x} - \boldsymbol{\mu}_i^t\| < \|\mathbf{x} - \boldsymbol{\mu}_j^t\|\right]$ is at most $\exp(-\Omega(C_{ij}^2))$, their overall contribution is at most $\exp(-\Omega(C_{ij}^2))d\sigma^2$. That is,

$$\Pr_{\mathbf{x}}\left[\|\mathbf{x} - \boldsymbol{\mu}_i^t\| < \|\mathbf{x} - \boldsymbol{\mu}_j^t\|\right] \cdot \mathbb{E}\left[\|(I - \mathbf{a}\mathbf{a}^T)\mathbf{z}\|^2 \,\Big|\, \|\mathbf{x} - \boldsymbol{\mu}_i^t\| < \|\mathbf{x} - \boldsymbol{\mu}_j^t\|\right] \le \exp(-\Omega(C_{ij}^2))d\sigma^2.$$

So finally it remains to bound $\mathbb{E}\left[\langle \mathbf{z}, \mathbf{a}\rangle^2 \,|\, \langle \mathbf{z}, \mathbf{a}\rangle < \tau\right]$. To this end, define the random variable $\hat{z} \equiv \langle \mathbf{z}, \mathbf{a}\rangle$. So we simply need to to upper bound $Pr\left[\hat{z} < \tau\right]\mathbb{E}\left[\hat{z}^2 | \hat{z} < \tau\right]$. But now note that $Pr\left[\hat{z} < \tau\right] \le \exp(-\Omega(C_{ij})^2)$. So using standard calculus, we get that this quantity attains a maximum value of $O(C_{ij}^2\exp(-\Omega(C_{ij}^2)))\sigma^2$. This completes the proof. $\qquad\square$

**Lemma 7.** *Suppose $\mathcal{I}_t$ holds, and fix any $i, j \ne i$. Then, if $\mathbf{x} \sim \mathrm{Cl}(j)$, we have* $\Pr_{\mathbf{x}}\left[g_i^t(\mathbf{x}) = 1\right]\mathbb{E}\left[\langle \boldsymbol{\mu}_i^t - \boldsymbol{\mu}_i^\star, \mathbf{x} - \boldsymbol{\mu}_i^\star\rangle \,\Big|\, g_i^t(\mathbf{x}) = 1\right] \le \frac{\widetilde{V}_t}{2k^5} + O(1)\exp(-C^2/8)(C^2 + k)\sigma^2.$

*Proof.* Again we provide an intuitive proof sketch before giving the formal proof. We first upper bound the inner product inside the expectation by $O(1)(\|\mathbf{x} - \boldsymbol{\mu}_j^\star\|^2 + \|\boldsymbol{\mu}_i^\star - \boldsymbol{\mu}_j^\star\|^2 + \|\boldsymbol{\mu}_i^t - \boldsymbol{\mu}_i^\star\|^2)$. On average, the first term is $d\sigma^2$, the second term is $C_{ij}^2\sigma^2$, and the third is at most $\widetilde{V}_t$, the maximum squared cluster error at time $t$. But note now that this quantity is multiplied by the probability $\Pr_{\mathbf{x}}\left[g_i^t(\mathbf{x}) = 1\right]$, which is at most $\Pr_{\mathbf{x}}\left[\|\mathbf{x} - \boldsymbol{\mu}_i^t\| < \|\mathbf{x} - \boldsymbol{\mu}_j^t\|\right] \le \exp(-\Omega(C_{ij}^2))$.

So now we consider two cases: in the first case, $\widetilde{V}_t \ge \sigma^2/C_{ij}^2$. Here, we can charge all the three terms in terms of $\widetilde{V}_t$, and since $C \ge \Omega(\sqrt{\log k})$, the overall expression can be bounded by $\frac{\widetilde{V}_t}{2k^5}$; in the second case, $\widetilde{V}_t < \sigma^2/C_{ij}^2$. Here, we can use Lemma 2 instead of Lemma 1 to get a much more accurate failure probability of $\exp(-C_{ij}^2/8)$. So we now get a bound of $O(1)\exp(-C^2/8)(C^2 + d)\sigma^2$. Combining the two cases yields the lemma. We can improve upon the $d\sigma^2$ term with a more careful analysis.

**Detailed Proof**: Again, let $\mathbf{x} = \boldsymbol{\mu}_j^\star + \mathbf{z}$ where $\mathbf{z}$ is sampled from a spherical normal Gaussian with mean $0$ and variance $\sigma^2$ along each direction. Firstly, note that:

$$\mathbb{E}\left[\langle \boldsymbol{\mu}_i^t - \boldsymbol{\mu}_i^\star, (\boldsymbol{\mu}_j^\star + \mathbf{z} - \boldsymbol{\mu}_i^\star)\rangle \,\Big|\, \mathcal{I}_t \text{ and } g_i^t(\boldsymbol{\mu}_{\mathbf{j}}^\star + \mathbf{z}) = 1\right] \le \tfrac{1}{2}\left(\|\boldsymbol{\mu}_i^t - \boldsymbol{\mu}_i^\star\|^2 + \|\boldsymbol{\mu}_j^\star - \boldsymbol{\mu}_i^\star\|^2\right)$$
$$+ \mathbb{E}\left[\langle \boldsymbol{\mu}_i^t - \boldsymbol{\mu}_i^\star, \mathbf{z}\rangle \,\Big|\, \mathcal{I}_t \text{ and } g_i^t(\boldsymbol{\mu}_{\mathbf{j}}^\star + \mathbf{z}) = 1\right].$$

Now, note that $g_i^t(\mathbf{x}) = 1$ depends on only the projection of $\mathbf{x}$ onto the subspace spanned by the $k+1$ vectors $\boldsymbol{\mu}_j^\star$ and the candidate centers $\boldsymbol{\mu}_{j'}^t$ for all $j'$. So the projection of $\mathbf{z}$ on directions orthogonal to this subspace remain normal variables with mean 0 and variance $\sigma^2$, and hence their contribution to the inner product is 0. So we effectively only need to bound:

$$\mathbb{E}\left[\left\langle \boldsymbol{\mu}_i^t - \boldsymbol{\mu}_i^\star, \mathbf{z}_\Pi \right\rangle \Big| \mathcal{I}_t \text{ and } g_i^t(\boldsymbol{\mu}_\mathbf{j}^\star + \mathbf{z}) = 1 \right]$$
$$\leq \frac{1}{2}\|\boldsymbol{\mu}_i^t - \boldsymbol{\mu}_i^\star\|^2 + \frac{1}{2}\mathbb{E}\left[\|\mathbf{z}_\Pi\|^2 \Big| \mathcal{I}_t \text{ and } g_i^t(\boldsymbol{\mu}_\mathbf{j}^\star + \mathbf{z}) = 1 \right].$$

So overall, the quantity we are seeking to bound in the Lemma is at most:

$$\Pr_{\mathbf{x} \sim \mathrm{Cl}(j)}\left[ g_i^t(\mathbf{x}) = 1 \Big| \mathcal{I}_t \right]$$
$$\cdot \left( \|\boldsymbol{\mu}_i^t - \boldsymbol{\mu}_i^\star\|^2 + \frac{1}{2}\|\boldsymbol{\mu}_i^\star - \boldsymbol{\mu}_j^\star\|^2 + \mathbb{E}\left[\|\mathbf{z}_\Pi\|^2 \Big| \mathcal{I}_t \text{ and } g_i^t(\boldsymbol{\mu}_\mathbf{j}^\star + \mathbf{z}) = 1 \right] \right).$$

Finally, it is easy to see that this expression is at most:

$$\Pr_{\mathbf{x} \sim \mathrm{Cl}(j)}\left[ \|\mathbf{x} - \boldsymbol{\mu}_i^t\| < \|\mathbf{x} - \boldsymbol{\mu}_j^t\| \Big| \mathcal{I}_t \right]$$
$$\cdot \left( \|\boldsymbol{\mu}_i^t - \boldsymbol{\mu}_i^\star\|^2 + \frac{1}{2}\|\boldsymbol{\mu}_i^\star - \boldsymbol{\mu}_j^\star\|^2 + \mathbb{E}\left[\|\mathbf{z}_\Pi\|^2 \Big| \mathcal{I}_t \text{ and } \|\mathbf{x} - \boldsymbol{\mu}_i^t\| < \|\mathbf{x} - \boldsymbol{\mu}_j^t\| \right] \right).$$

We now consider two cases, depending on whether $\max(\|\boldsymbol{\mu}_i^t - \boldsymbol{\mu}_i^\star\|, \|\boldsymbol{\mu}_j^t - \boldsymbol{\mu}_j^\star\|) > \sigma/C_{ij}$ or not.

**Case (i):** $\max(\|\boldsymbol{\mu}_i^t - \boldsymbol{\mu}_i^\star\|, \|\boldsymbol{\mu}_j^t - \boldsymbol{\mu}_j^\star\|) > \sigma/C_{ij}$. In this case we will show that the desired expression is at most $\frac{\widetilde{V}_t}{k^5}$. Indeed, to this end, firstly note that we have $\|\boldsymbol{\mu}_i^\star - \boldsymbol{\mu}_j^\star\|^2 = C_{ij}^2\sigma^2 \leq C_{ij}^4\widetilde{V}_t$. Moreover, note that from Lemma 1 we have:

$$\Pr_{\mathbf{x} \sim \mathrm{Cl}(j)}\left[ \|\mathbf{x} - \boldsymbol{\mu}_i^t\| < \|\mathbf{x} - \boldsymbol{\mu}_j^t\| \Big| \mathcal{I}_t \right] \leq \exp(-\Omega(C_{ij})^2).$$

So using this, we can show in a manner akin to Lemma 6 that $\Pr_{\mathbf{x} \sim \mathrm{Cl}(j)}\left[ \|\mathbf{x} - \boldsymbol{\mu}_i^t\| < \|\mathbf{x} - \boldsymbol{\mu}_j^t\| \Big| \mathcal{I}_t \right] \mathbb{E}\left[\|\mathbf{z}_\Pi\|^2 \Big| \mathcal{I}_t \text{ and } \|\mathbf{x} - \boldsymbol{\mu}_i^t\| < \|\mathbf{x} - \boldsymbol{\mu}_j^t\| \right]$ is at most $O(1)\exp(-\Omega(C_{ij})^2)(C_{ij}^2 + k)\sigma^2 \leq \exp(-\Omega(C_{ij})^2)(C_{ij}^2 + k)C_{ij}^2\widetilde{V}_t$. Putting everything together, we get that the desired quantity we need to bound is at most $O(1)\exp(-\Omega(C_{ij})^2)\widetilde{V}_t\left(1 + C_{ij}^4 + + kC_{ij}^2\right)$. Overall this is at most $\frac{\widetilde{V}_t}{2k^5}$ since $C_{ij} \geq C = \Omega(\sqrt{\log k})$.

**Case (ii):** $\max(\|\boldsymbol{\mu}_i^t - \boldsymbol{\mu}_i^\star\|, \|\boldsymbol{\mu}_j^t - \boldsymbol{\mu}_j^\star\|) < \sigma/C_{ij}$. In this case, we want to replace the use of Lemma 1 with Lemma 2 in the above proof. Indeed, from Lemma 2 we have:

$$\Pr_{\mathbf{x} \sim \mathrm{Cl}(j)}\left[ \|\mathbf{x} - \boldsymbol{\mu}_i^t\| < \|\mathbf{x} - \boldsymbol{\mu}_j^t\| \Big| \mathcal{I}_t \right] \leq \exp(-C_{ij}^2/8).$$

So using this, we can show in a manner akin to Lemma 6 that:

$$\Pr_{\mathbf{x} \sim \mathrm{Cl}(j)}\left[ \|\mathbf{x} - \boldsymbol{\mu}_i^t\| < \|\mathbf{x} - \boldsymbol{\mu}_j^t\| \Big| \mathcal{I}_t \right] \mathbb{E}\left[\|\mathbf{z}_\Pi\|^2 \Big| \mathcal{I}_t \text{ and } \|\mathbf{x} - \boldsymbol{\mu}_i^t\| < \|\mathbf{x} - \boldsymbol{\mu}_j^t\| \right]$$
$$\leq O(1)\exp(-C_{ij}^2/8)(C_{ij}^2 + k)\sigma^2.$$

Putting everything together, we get that the desired quantity we need to bound in the Lemma statement is at most $O(1)\exp(-C_{ij}/8^2)\sigma^2\left(C_{ij}^2 + k\right)$. $\qquad \square$

**Lemma 8.** *Suppose $\mathcal{I}_t$ holds. For any $i$ let $\mathbf{x} \sim \mathrm{Cl}(i)$. Then we have:*

$$\Pr_{\mathbf{x}}\left[g_i^t(\mathbf{x}) = 0\right] \mathbb{E}\left[\left\langle \boldsymbol{\mu}_i^t - \boldsymbol{\mu}_i^\star, \mathbf{x} - \boldsymbol{\mu}_i^\star \right\rangle \Big| g_i^t(\mathbf{x}) = 0 \right] \geq -\frac{\widetilde{V}_t}{2k^5} - O(k)\exp(-C^2/8)(C^2 + k)\sigma^2.$$

*Proof.* Note that the expectation above is exactly:

$$\mathbb{E}\left[\left\langle \boldsymbol{\mu}_i^t - \boldsymbol{\mu}_i^\star, (\boldsymbol{\mu}_i^\star + \mathbf{z} - \boldsymbol{\mu}_i^\star)\right\rangle \,\Big|\, g_t^t(\boldsymbol{\mu}_{\mathbf{i}}^\star + \mathbf{z}) = 0\right] = \mathbb{E}\left[\left\langle \boldsymbol{\mu}_i^t - \boldsymbol{\mu}_i^\star, \mathbf{z}\right\rangle \,\Big|\, g_i^t(\boldsymbol{\mu}_{\mathbf{i}}^\star + \mathbf{z}) = 0\right].$$

Now, note that the condition $g_i^t(\mathbf{x}) = 0$ depends on only the projection of $\mathbf{x}$ onto the subspace spanned by the $k + 1$ vectors $\boldsymbol{\mu}_i^\star$ and the candidate centers $\boldsymbol{\mu}_{j'}^t$, for all $j'$. So the projection of $\mathbf{z}$ on directions orthogonal to this subspace remain normal variables with mean 0 and variance $\sigma^2$, and hence their contribution to the inner product is 0. So we effectively only need to bound $\mathbb{E}\left[\left\langle \boldsymbol{\mu}_i^t - \boldsymbol{\mu}_i^\star, \mathbf{z}_\Pi\right\rangle \,\Big|\, \mathcal{I}_t \text{ and } g_i^t(\boldsymbol{\mu}_{\mathbf{i}}^\star + \mathbf{z}) = 0\right]$. Here, $\mathbf{z}_\Pi$ denotes the projection of $\mathbf{z}$ onto the subspace spanned by the $k$ candidate centers and the true mean $\boldsymbol{\mu}_j^\star$. So overall, we get the following:

$$\mathbb{E}\left[\left\langle \boldsymbol{\mu}_i^t - \boldsymbol{\mu}_i^\star, \mathbf{z}\right\rangle \,\Big|\, g_t^t(\boldsymbol{\mu}_{\mathbf{i}}^\star + \mathbf{z}) = 1\right] \geq -\frac{1}{2}\left(\|\boldsymbol{\mu}_i^t - \boldsymbol{\mu}_i^\star\|^2 + \mathbb{E}\left[\|\mathbf{z}_\Pi\|^2 \,\Big|\, g_i^t(\boldsymbol{\mu}_{\mathbf{i}}^\star + \mathbf{z}) = 0\right]\right).$$

Again, ignoring the effect of conditioning — we deal with this in a manner identical to that in the proof of Lemma 6—the LHS above is at least $-\frac{1}{2}\left(\widetilde{V}_t + k\sigma^2\right)$. Finally we note that $\Pr_{\mathbf{x} \sim \mathrm{Cl}(i)}[g_i^t(\mathbf{x}) = 0]$ is at most $\sum_{j \neq i} \Pr_{\mathbf{x} \sim \mathrm{Cl}(i)}\left[\|\mathbf{x} - \boldsymbol{\mu}_i^t\| \geq \|\mathbf{x} - \boldsymbol{\mu}_j^t\|\right]$, which in turn is at most $k \exp(-\Omega(C^2))$ by Lemma 1. So multiplying these, and using two cases similar to the above proof, we get that the overall expression is at least $-\frac{\widetilde{V}_t}{2k^5} - O(k)\exp(-C^2/8)\left(C^2 + k\right)\sigma^2$ as long as $C = \Omega(\sqrt{\log k})$. □

## B Complete Details: Ensuring Proximity Condition Via Super-Martingales

Our next key result is to show that a sample path through $N$ steps satisfies $\mathcal{I}_N = 1$ w.p $\geq 1 - \frac{1}{\mathrm{poly(N)}}$, for suitable initialization. Recall that $\mathcal{I}_N = 1$ if that $\max_i \|\boldsymbol{\mu}_i^0 - \boldsymbol{\mu}_i^\star\| \leq \frac{C\sigma}{10}$ for all $0 \leq t \leq N$. In the rest of this section, we assume that the center separation $C \geq \Omega(\sqrt{\log k})$.

**Theorem 8.** *Suppose our initial estimates $\boldsymbol{\mu}_i^0$ satisfy $\max_i \|\boldsymbol{\mu}_i^0 - \boldsymbol{\mu}_i^\star\| \leq \frac{C\sigma}{20}$, then w.p $\geq 1 - \frac{1}{\mathrm{poly(N)}}, \mathcal{I}_t = 1 \,\forall\, 1 \leq t \leq N$.*

*Proof.* We first recall and define some useful quantities. Firstly, $\tilde{E}_t^i = \|\boldsymbol{\mu}_i^t - \boldsymbol{\mu}_i^\star\|^2$ denotes the current squared error for cluster $i$ after the $t^{th}$ streaming update. It's also useful to define the quantity $e_t^i = \|\boldsymbol{\mu}_i^t - \boldsymbol{\mu}_i^\star\|$. Another important quantity is $\tilde{V}_t = \max_i \tilde{E}_t^i$, and analogously $v^t = \max_i e_t^i$. We will repeatedly use that $\tilde{V}^t = (v^t)^2$ and $\tilde{E}_t^i = (e_t^i)^2$.

Our argument proceeds as follows. Intuitively, if we start at $\frac{C\sigma}{20}$ within the true clusters, we need a lot of *bad events* to keep moving the estimates away and out of the $\frac{C\sigma}{20}$ radius ball. Since the samples $\mathbf{x^t}$ are independent, we can use a concentration inequality to bound the probability with which a lot of bad events occur together. At a high level, we can think of $Z(t) = e_{t+1}^i - e_t^i$ as a martingale difference sequence on which we want to apply a concentration inequality. In order to do so, we perform the following steps.

- Bound $\mathbb{E}[Z(t) \mid \mathbf{x^1}, \mathbf{x^2}, \ldots \mathbf{x^t}]$, and show conditions under which this quantity is negative (since we want to show decrease). This is discussed in Lemma 9. At a high level, we show that the error term decreases on average in one step if the current error is sufficiently large (i.e., at least $C\sigma/20$). So we start a super-martingale series whenever the error term exceeds this value, and show that the probability of this series exceeding $C\sigma/10$ is negligible. We stop this series if the error falls below $C\sigma/20$, and start a new series when the error next exceeds this lower threshold of $C\sigma/20$. Since there can be at most $Nk$ such series' (across clusters), a simple union bound would then suffice.

- While the differences $Z(t)$ are not bounded, they have sub-gaussian tails. When a point is correctly clustered, the error is roughly the norm of a Gaussian variable and is hence sub-gaussian. When a point is incorrectly classified, the error is sub-gaussian. However the the mean is approximately $C_{ij}\sigma$, which can be arbitrarily large. We deal with this by using Lemma 1 which says that the probability of misclassification is small. There are some more technical details which are covered in Lemma 11.

- As a next step, we use Azuma Hoeffding style inequality for sub-gaussians [15] to complete the proof (Lemma 14). However, we need to be careful while defining the martingale sequences on which we apply this concentration inequality in order to satisfy the conditions required in Lemma 9 and Lemma 11. This forms the final part of the proof.

**Lemma 9.** *When* $\frac{C\sigma}{20} \leq e_t^i$ *and* $v^t \leq \frac{C\sigma}{10}$, *we have* $\mathbb{E}_{\mathbf{x}^{t+1}}[e_{t+1}^i] \leq e_t^i$.

*Proof.* We first show that $\mathbb{E}_{\mathbf{x}^{t+1}}[\tilde{E}_{t+1}^i] \leq \tilde{E}_t^i$. The proof follows directly from Lemma 4 (reproduced below)

$$\mathbb{E}_{\mathbf{x}^{t+1}}[\tilde{E}_{t+1}^i] \leq (1 - \frac{\eta}{3k})\tilde{E}_t^i + \underbrace{\frac{\eta}{k^5}\tilde{V}_t + O(1)\eta^2 d\sigma^2}_{f(C,\eta,k)} + \underbrace{O(k)\eta(1-\eta)\exp(-C^2)C^2\sigma^2}_{g(C,\eta,k)} .$$

Setting $\eta = \frac{3k\log 3N}{N}$. The term $f(C,\eta,k) + g(C,\eta,k) \leq \frac{\eta}{3k}\tilde{E}_t^i$ when $\tilde{V}^t \leq \frac{C^2\sigma^2}{100}$ and $\tilde{E}_t^i \geq \frac{C^2\sigma^2}{400}$.

Therefore, $\mathbb{E}_{\mathbf{x}}\left[\tilde{E}_{t+1}^t\right] \leq \tilde{E}_t^i$. By Jensen's inequality, we have $(\mathbb{E}_{\mathbf{x}^{t+1}}[e_{t+1}^i])^2 \leq \mathbb{E}_{\mathbf{x}^{t+1}}[\tilde{E}_{t+1}^i]$, which in turn we showed is $\leq \tilde{E}_t^i$. Taking square-roots, we get the required result. $\square$

In order to bound the deviation from the mean, we appeal to Azuma style inequality for Subgaussians [15]. We show next that $e_{t+1}^i - e_t^i$ has sub-gassian behaviour under some conditions. Since $\mathbb{E}_{\mathbf{x}^{t+1}}[e_i^{t+1}] - e_i^t$ is not zero, we'd need the following lemma to deal with tail behaviour for non-zero mean variables.

**Lemma 10.** *Suppose $X$ is a random variable that satisfies* $\Pr[X \geq a] \leq b_0 \exp\left(\frac{-a^2}{\sigma_0^2}\right)$ *for some* $b_0 \geq 1$. *Then for any $\delta > 0$,*

$$\Pr[X + \delta \geq a] \leq b_0 \exp\left(\frac{\delta^2}{4C^2\sigma_0^2}\right)\exp\left(\frac{-a^2}{4C^2\sigma_0^2}\right).$$

*Similarly, suppose we have* $\Pr(X \leq -a) \leq b_0 \exp\left(\frac{-a^2}{\sigma_0^2}\right)$ *for some $b_0 \geq 1$, then for any $\delta > 0$,*

$$\Pr[X - \delta \leq -a] \leq b_0 \exp\left(\frac{\delta^2}{4C^2\sigma_0^2}\right)\exp\left(\frac{-a^2}{4C^2\sigma_0^2}\right).$$

*Proof.* For $a \geq \delta$, we have $\Pr[X + \delta \geq a] = \Pr[X \geq (a - \delta)] = \exp\left(\frac{-(a-\delta)^2}{\sigma_0^2}\right)$.

Hence we need to essentially show that $(a - \delta)^2 \geq \frac{a^2}{4C^2} - \frac{\delta^2}{4C^2}$.

When $a \geq \frac{2\delta c}{2c-1}$, we have $(a - \delta)^2 \geq \frac{a^2}{4C^2}$.

In the range, $\delta \leq a \leq \frac{2\delta c}{2c-1}$ the quadratic $(a - \delta)^2 + \frac{\delta^2}{4C^2} - \frac{a^2}{4C^2} \geq 0$. This can be verified by noting that $a = \delta$ is the only root in the interval and the condition is true at the end points of the interval.

An identical proof holds for bounding $\Pr[X - \delta \leq -a]$ when $a \geq \delta$.

When $a \leq \delta$, the quantity $b_0 \exp\left(\frac{\delta^2 - a^2}{4C^2\sigma_0^2}\right)$ is greater than 1 and hence the result is trivially true in this case. $\square$

We now study the tail behaviour of the quantity $e_{t+1}^i - e_t^i$.

**Lemma 11.** *The random variable* $d_t^i = e_{t+1}^i - e_t^i$ *has sub-gaussian tails, when* $v^t \leq \frac{C\sigma}{10}$. *More precisely,*

$$\forall a \geq 0, \Pr[d_t^i \geq a] \leq \exp\left(-\frac{a^2}{\eta^2\sigma^2 d}\right), \text{ and}$$

$$\forall a \geq 0, \Pr[d_t^i \leq -a] \leq \exp\left(-\frac{a^2}{\eta^2C^2\sigma^2 d}\right)\exp\left(\frac{1}{100}\right).$$

*Proof.* For brevity in notation, we denote the random variable $\mathbf{x^{t+1}}$ as $\mathbf{x}$. Also, let $\mathbf{w} = g_i^t(\mathbf{x})\mathbf{x} + (1 - g_i^t(\mathbf{x}))\boldsymbol{\mu}_i^t$, and let $\mathbf{z}$ denote a zero mean Gaussian in $d$ dimensions with variance $\sigma^2 I$.

We bound the probability that $d_t^i \geq a$, for some $a > 0$. We work through this in cases.

**Case i.** Let $\mathbf{x} \sim Cluster(j), j \neq i$. Recall from the update rule (Equation 4.1),

$$e_{t+1}^i - e_t^i = \|(1-\eta)\boldsymbol{\mu}_i^t + \eta\mathbf{w} - \boldsymbol{\mu_i^\star}\| - \|\boldsymbol{\mu_i^t} - \boldsymbol{\mu_i^\star}\|$$
$$\leq \eta\|\mathbf{w} - \boldsymbol{\mu_i^t}\|.$$

For convenience, let's consider the random variable $B = \|\mathbf{w} - \boldsymbol{\mu_i^t}\|$ and bound $p = \Pr[B \geq a']$.

Clearly, this term is 0 when $g_i^t(\mathbf{x}) = 0$, and $\|\mathbf{x} - \boldsymbol{\mu}_i^t\|$ otherwise. Therefore, $\Pr[B \geq a'] \leq \underbrace{\Pr[g_i^t(\mathbf{x}) = 1]}_{p_1}$ and $\Pr[B \geq a'] \leq \underbrace{\Pr[\|\mathbf{x} - \boldsymbol{\mu}_i^t\| \geq a']}_{p_2}$.

We can bound $p_2$ using Standard Gaussian concentration. Let $\mathbf{x} = \boldsymbol{\mu}_j^\star + \mathbf{z}$. Then $\|\mathbf{x} - \boldsymbol{\mu}_i^t\| \leq \|\mathbf{z}\| + \frac{C\sigma}{10} + C_{ij}\sigma \leq \|\mathbf{z}\| + \frac{11}{10}C_{ij}\sigma$.

Therefore, $p_2 \leq \exp\left(-\frac{(a' - \frac{11}{10}C_{ij}\sigma)^2}{\sigma^2 d}\right)$.

Hence, when $a' \geq \Omega(C_{ij})\sigma$, we have $p \leq p_2 \leq \exp(-\frac{a'^2}{2\sigma^2 d})$. From Lemma 1, when $a' \leq \Omega(C_{ij})\sigma$, $p \leq p_1 = \exp(-\Omega(C_{ij}^2)) \leq \exp(-\frac{a'^2}{\sigma^2})$.

The random variable of interest, $d_i^t = \eta B$. Therefore, $\Pr[d_i^t \geq a] \leq \exp\left(-\frac{a^2}{\eta^2\sigma^2 d}\right)$.

**Case ii.** Let $\mathbf{x} \sim Cluster(i)$. Just as before, if $g_i^t(\mathbf{x}) = 0$, $A = 0$. However in the case that $g_i^t(\mathbf{x}) = 1$, we bound $d_i^t$ slightly differently.

$$d_i^t = \|(1-\eta)\boldsymbol{\mu}_i^t + \eta\mathbf{x} - \boldsymbol{\mu}_i^\star\| - \|\boldsymbol{\mu}_i^t - \boldsymbol{\mu}_i^\star\|$$
$$\leq -\eta\|\boldsymbol{\mu}_i^t - \boldsymbol{\mu}_i^\star\| + \eta\|\mathbf{x} - \boldsymbol{\mu_i^\star}\|$$
$$\leq \eta\|\mathbf{z}\|,$$

where the last inequality follows from $\mathbf{x} = \boldsymbol{\mu}_i^\star + \mathbf{z}$ in this case. $\Pr[A \geq a] \leq \Pr[\|\mathbf{z}\| \geq \frac{a}{\eta}] \leq \exp\left(-\frac{a^2}{\eta^2\sigma^2 d}\right)$ by standard Gaussian concentration.

We now look at bounding the negative tails. Just as before, we consider two cases.

**Case i.** Let $\mathbf{x} \sim Cluster(j), j \neq i$. Recall from the update rule (Equation 4.1),

$$e_{t+1}^i - e_t^i = \|(1-\eta)\boldsymbol{\mu}_i^t + \eta\mathbf{w} - \boldsymbol{\mu}_i^\star\| - \|\boldsymbol{\mu}_i^t - \boldsymbol{\mu}_i^\star\|$$
$$\geq -\eta\|\mathbf{w} - \boldsymbol{\mu}_i^t\|.$$

Therefore, $\Pr[A \leq -a] = \Pr[B \geq \frac{a}{\eta}]$. This is the same quantity that we bounded for the positive tails.

**Case ii.** Let $\mathbf{x} \sim Cluster(i)$. In this case when $g_i^t(\mathbf{x}) = 1$, we get

$$e_{t+1}^i - e_t^i = \|(1-\eta)\boldsymbol{\mu}_i^t + \eta\mathbf{z}\| - \|\boldsymbol{\mu}_i^t - \boldsymbol{\mu}_i^\star\|$$
$$\geq -\eta\|\boldsymbol{\mu}_i^t - \boldsymbol{\mu}_i^\star\| - \eta\|\mathbf{z}\|$$
$$\geq -\eta\frac{C\sigma}{10} - \eta\|\mathbf{z}\|,$$

where the last inequality follows from the assumptions of the theorem. $\Pr[-\eta\|\mathbf{z}\| \leq -a] \leq \exp\left(\frac{a^2}{\eta^2\sigma^2 d}\right)$. Using Lemma 10 with $\delta = \frac{\eta C\sigma}{10}$, for $a \geq 0$, we get $\Pr[A \leq -a] \leq \exp\left(\frac{1}{400d}\right)\exp\left(-\frac{a^2}{4\eta^2 C^2\sigma^2 d}\right)$. The expected error can decrease a lot when the point is correctly classified, so we lose a factor in the sub-gaussian parameter while bounding the negative tails. $\square$

From the above lemma, we obtain a useful corollary which is stated below.

**Lemma 12.** *If $e^i_t \leq \frac{C\sigma}{20}$, then with probability $1 - \frac{1}{poly(N)}$ $e^i_{t+1} \leq \frac{3C\sigma}{40}$.*

*Proof.* Result follows from Lemma 11 with setting $a = \frac{C\sigma}{40}$ and $\eta = \frac{3k \log N}{N}$. $\square$

Equipped with lemmas 9 and 11, we can now put things together. We need to carefully handle the conditions under which the above lemmas hold. This motivates the definition of the following random processes.

$A^i(t) = 1$ if $e^t_i \geq \frac{C\sigma}{20}$ and $e^{t-1}_i < \frac{C\sigma}{20}$. In other words, $A^i(t)$ is set to 1 when $e^t_i$ crosses the threshold of $\frac{C\sigma}{20}$ from below.

We also define $Z^i_\tau$ as follows.

$$
Z^i_\tau(t) = \begin{cases} e^i_{t+1} - e^i_t \text{ if } \sum\limits_{t'=1}^{t} A(t') = \tau \text{ and } v^t < \frac{C\sigma}{10}, \\ 0, \text{ Otherwise} \end{cases}
$$

In other words, $Z^i_\tau$ is "active" and tracks the difference in error with time, when $\tau$ is the first time that error crosses $\frac{C\sigma}{20}$ from below and has not crossed $\frac{C\sigma}{10}$. It takes the value 0 at all other times.

**Lemma 13.** *For the sequence $Z^i_\tau(t) - \mathbb{E}_{\mathbf{x^{t+1}}}[Z^i_\tau(t)]$, there are constants $b > 1, c > 0$ such that for all $\tau, i, t$ and any $a \geq 0$, it holds that*

$$
\Pr(Z^i_\tau(t) - \mathbb{E}_{\mathbf{x^{t+1}}}[Z^i_\tau(t)] > a \mid \mathbf{x^1}, \ldots \mathbf{x^t}) \leq b \exp(-ca^2),
$$
$$
\Pr(Z^i_\tau(t) - \mathbb{E}_{\mathbf{x^{t+1}}}[Z^i_\tau(t)] < -a \mid \mathbf{x^1}, \ldots \mathbf{x^t}) \leq b \exp(-ca^2).
$$

*Proof.* The proof is mainly based on Lemma 11. Recall from the definition of $Z^i_\tau$ that $Z^i_\tau(t) = 0$ if $e^i_t < \frac{C\sigma}{20}$ or $v^t > \frac{C\sigma}{10}$. The lemma is trivially true in this case. Therefore, in the rest of the proof we assume that $\frac{C\sigma}{20} \leq e^i_i$ and $v^t \leq \frac{C\sigma}{10}$.

We know that $\mathbb{E}_{\mathbf{x^{t+1}}}[Z^i_\tau(t)] < 0$ (From Lemma 9).

However, in order to study the tail behaviour of $Z^i_\tau(t) - \mathbb{E}_{\mathbf{x^{t+1}}}[Z^i_\tau(t)]$ we need to also lower bound the quantity $\mathbb{E}_{\mathbf{x^{t+1}}}[Z^i_\tau(t)]$. Once again, we are only interested in the case that $\frac{C\sigma}{20} \leq e^t_i$ and $v^t \leq \frac{C\sigma}{10}$ and $Z^i_\tau(t) = e^i_{t+1} - e^i_t = d^i_t$. From the update rule, we get

$$
\begin{aligned}
\mathbb{E}[d^i_t] &= \|(\boldsymbol{\mu}^t_i - \boldsymbol{\mu}^\star_i) + \eta g(\mathbf{x})[\mathbf{x} - \boldsymbol{\mu^t_i}]\| - \|\boldsymbol{\mu^t_i} - \boldsymbol{\mu^\star_i}\| \\
&\geq -\eta \mathbb{E}[g(\mathbf{x})\|\mathbf{x} - \boldsymbol{\mu^\star_i}\|] - \eta\|\boldsymbol{\mu^\star_i} - \boldsymbol{\mu^t_i}\| \\
&\geq -\eta O(\sqrt{d})\sigma - \eta\frac{C\sigma}{10},
\end{aligned}
$$

where the last inequality follows from Lemma 4 and Jensen's inequality.

For ease of notation, let $\Delta = -\mathbb{E}[d^i_t] \leq \eta O(\sqrt{d})\sigma + \eta\frac{C\sigma}{10}$. Since we've now bounded the mean, we can use Lemma 11 and Lemma 10 to complete the proof as presented below.

**Positive tail.** Lemma 10 with $\delta = \Delta$, we get $\Pr[Z^i_\tau(t) - \mathbb{E}[Z^i_\tau(t)] \geq a] \leq O(1) \exp\left(-\frac{a^2}{4C^2\eta^2\sigma^2 d}\right)$, for $a \geq 0$.

**Negative tail.** $\Pr[Z^i_\tau(t) - \mathbb{E}[Z^i_\tau(t)] \leq -a] \leq \Pr[Z^i_\tau(t) \leq -(a + \Delta)] \leq \Pr[Z^i_\tau(t) \leq -a]$. From Lemma 11, $\forall a \geq 0$, we have $\Pr[Z^i_\tau(t) - \mathbb{E}[Z^i_\tau(t)] \geq a] \leq O(1) \exp\left(-\frac{a^2}{4C^2\eta^2\sigma^2 d}\right)$. $\square$

**Lemma 14.** *For all $\tau = 1, 2, \ldots N$ and $i = 1, 2, \ldots k$, with probability $1 - \delta$,*

$$
\sum_{t=0}^{N-1} Z^i_\tau(t) \leq \sqrt{\frac{\alpha C^2 \sigma^2 k^2 \log^2 N \log(\frac{1}{\delta})}{N}}, \tag{9}
$$

*where $\alpha$ is some constant.*

In particular, this gives us that $\Pr\left[\sum_{t=0}^{N-1} Z_\tau^i(t) \geq \frac{C\sigma}{40}\right] \leq \frac{1}{\text{poly}(N)}$.

*Proof.* We apply Azuma style concentration inequality to $Z_\tau^i(t) - \mathbb{E}_{\mathbf{x}^{t+1}}[Z_\tau^i(t)]$.

By definition, when $Z_\tau^i \neq 0$, the conditions for Lemma 9 and 11 apply. Hence we can apply Theorem 2 of [15] with parameters $b = O(1)$ and $c = \frac{1}{C^2 d\sigma^4 \eta^2}$. From Lemma 9, $\mathbb{E}[Z_\tau^i] \leq 0$ and hence the result. $\qquad\square$

We say that a process $Z_\tau^i$ fails if there is a time $T$ such that $e_i^t > \frac{C\sigma}{10}$ and $Z_\tau^i(T) = e_t^{T+1} - e_i^T$, *i.e.* $Z_\tau^i$ was active at the instant $T$. By definition of $Z_\tau^i$, $\sum_{i=1}^{N} Z_\tau^i(t) = e_{T+1}^i - e_\tau^i$.

Recall that $\tau$ is the first time that $Z_\tau^i$ crosses $\frac{C\sigma}{20}$ from below. Lemma 12 gives us that with probability $1 - \frac{1}{\text{poly}(N)}$, $e_\tau^i \leq \frac{3C\sigma}{40}$. Taking union bound over both the bad events of Lemma 12 and 14, we get that the probability that $Z_\tau^i$ fails is $\leq \frac{1}{\text{poly}(N)}$.

For $\mathcal{I}_t = 0$, atleast one of the process $Z_\tau^i$ has to fail (since we start with initialization below $\frac{C\sigma}{20}$ and can't fail in a single step by Lemma 12). Taking union bound over the $kN$ such processes gives us the required result. $\qquad\square$

## C  Complete Details: Soft streaming updates

We first restate our main result here.

**Theorem 9** (Streaming Update). *Let $\mathbf{x^t}$, $1 \leq t \leq N + N_0$ be generated using a mixture two balanced spherical Gaussians. Also, let the center-separation $C \geq 4$, and also suppose our initial estimate $\boldsymbol{\mu}^0$ is such that $\|\boldsymbol{\mu}^0 - \boldsymbol{\mu}^\star\| \leq \frac{C\sigma}{20}$.*

*Then, the streaming update of* StreamSoftUpdate$(N, N_0)$*, i.e, Steps 3-8 of Algorithm 3 satisfies:*

$$E_N \leq \underbrace{\frac{E_0}{N^{\Omega(1)}}}_{\text{bias}} + \underbrace{O(1)\frac{\log N}{N} d\sigma^2}_{\text{variance}},$$

*where $E_t = \mathbb{E}\left[\|\boldsymbol{\mu}^t - \boldsymbol{\mu}^\star\|^2\right]$.*

*Proof.* The proof for the soft streaming updates follows the same skeleton as that of streaming hard updates which is detailed in Section 4. We first bound the quantity $\widehat{E}_{t+1} = \mathbb{E}_{\mathbf{x^{t+1}}}\left[\|\boldsymbol{\mu}^{t+1} - \boldsymbol{\mu}^\star\|^2\right]$ (analogous to Lemma 4).

**Lemma 15.** *Suppose $\mathcal{I}_t$ is satisfied, i.e. $\|\boldsymbol{\mu}^t - \boldsymbol{\mu}^\star\| \leq \frac{C\sigma}{10}$ then,*

$$\widehat{E}_{t+1} \leq \left(1 - \frac{3\eta}{8}\right)\tilde{E}_t + O(1)\eta^2 d\sigma^2.$$

*Proof.* For brevity of notation, we refer to $\mathbf{x^{t+1}}$ as $\mathbf{x}$. All expectations are taken with respect to $\mathbf{x}$. Recall the update rule for soft streaming.

$$\boldsymbol{\mu}^{t+1} = (1 - \eta)\boldsymbol{\mu}^t + \eta[2w_t(\mathbf{x}) - 1]\mathbf{x}.$$

Expanding the expression for $\widehat{E}_{t+1}$, we get

$$\widehat{E}_{t+1} = \mathbb{E}\left[\|\boldsymbol{\mu}^{t+1} - \boldsymbol{\mu}^\star\|^2\right]$$
$$= (1 - \eta)^2 \tilde{E}_t + 2\eta(1 - \eta)\Big\langle \boldsymbol{\mu}^t \boldsymbol{\mu}^\star, \underbrace{\mathbb{E}\left[(2w^t(\mathbf{x}) - 1)\mathbf{x}\right]}_{\mathbf{y^t}} - \boldsymbol{\mu}^\star\Big\rangle +$$
$$\eta^2\|\boldsymbol{\mu}^\star\|^2 + \eta^2\mathbb{E}\left[(2w^t(\mathbf{x}) - 1)\|\mathbf{x}\|^\mathbf{2}\right] - 2\eta^2\Big\langle\boldsymbol{\mu}^\star, \underbrace{\mathbb{E}\left[(2w^t(\mathbf{x}) - 1)\mathbf{x}\right]}_{\mathbf{y^t}}\Big\rangle \qquad (10)$$

From the above expression, we can see that the key quantity to bound is $\mathbb{E}\left[(2w^t(\mathbf{x}) - 1)\mathbf{x}\right] \overset{\text{def}}{=} \mathbf{y^t}$. The next lemma provides an expression for $\mathbf{y^t}$.

**Lemma 16.** *Suppose $\mathcal{I}_t = 1$. We have the following expression for $\mathbf{y^t} = \mathbb{E}\left[(2w^t(\mathbf{x}) - 1)\mathbf{x}\right]$.*

$$\mathbf{y^t} = 2\gamma^t(\boldsymbol{\mu}^t - \boldsymbol{\mu}^\star) + \boldsymbol{\mu}^\star, \ \ where$$

$$\gamma^t \leq \frac{1}{8C^2},$$

*Proof.* For simplicity, let's define the following two terms.

$$\mathbf{y_1}^\mathbf{t} \overset{\text{def}}{=} \mathbb{E}\left[w^t(\mathbf{x})\mathbf{x} \mid \mathbf{x} \sim \mathcal{N}(\boldsymbol{\mu}^\star, \sigma^2 I)\right]$$

$$\mathbf{y_2}^\mathbf{t} \overset{\text{def}}{=} \mathbb{E}\left[w^t(\mathbf{x})\mathbf{x} \mid \mathbf{x} \sim \mathcal{N}(-\boldsymbol{\mu}^\star, \sigma^2 I)\right]$$

Note that $\mathbf{y^t} = \frac{1}{2}\left(\mathbf{y_1^t} + \mathbf{y_2^t}\right)$. Our first observation is that $\mathbf{y_2}^\mathbf{t}$ takes the form $\gamma(\boldsymbol{\mu}^t - \boldsymbol{\mu}^\star)$ for some $\gamma \in \mathbb{R}$.

$$\mathbf{y_2^t} = \int\limits_{-\infty}^{\infty} p(\mathbf{x})w^t(\mathbf{x})\mathbf{x}dx,$$

$$p(\mathbf{x})w(\mathbf{x}) = \frac{1}{\sqrt{2\pi\sigma^2}}\frac{\exp\left(\frac{-\|\mathbf{x}-\boldsymbol{\mu^t}\|^2 - \|\mathbf{x}+\boldsymbol{\mu}^\star\|^2}{\sigma^2}\right)}{\exp\left(\frac{-\|\mathbf{x}-\boldsymbol{\mu^t}\|^2}{\sigma^2}\right) + \exp\left(\frac{-\|\mathbf{x}+\boldsymbol{\mu^t}\|^2}{\sigma^2}\right)}.$$

When $\langle\boldsymbol{\mu}^t - \boldsymbol{\mu}^\star, x\rangle = 0$, we have $p(\mathbf{x})w(\mathbf{x}) = p(-\mathbf{x})w(-\mathbf{x})$. Therefore, the terms cancel in the integration and we get $T_2 = \gamma(\boldsymbol{\mu}^t - \boldsymbol{\mu}^\star)$ for some $\gamma \in \mathbb{R}$.

Our next observation is that $\mathbf{y_1^t} - \mathbf{y_2^t} = \boldsymbol{\mu}^\star$. This can be verified by seeing that for every term corresponding to $\boldsymbol{\mu}^\star + \mathbf{z}$ in $\mathbf{y_1^t}$, there is a term corresponding to $-\boldsymbol{\mu}^\star - \mathbf{z}$ in $\mathbf{y_2^t}$. These two terms have the same multiplier $p(\mathbf{x})$ but weight multipliers summing to one. Hence taking the difference of each term in $\mathbf{y_1^t}$ and $\mathbf{y_2^t}$ and integrating gives the required result. Since $\mathbf{y^t} = \frac{1}{2}\left(\mathbf{y_1^t} + \mathbf{y_2^t}\right)$, we get that $\mathbf{y^t} = 2\gamma^t(\boldsymbol{\mu}^t - \boldsymbol{\mu}^\star) + \boldsymbol{\mu}^\star$ where $\gamma^t$ is some constant.

We now bound the value of this constant $\gamma^t$. Theorem 2 of [6] gives the following bound on $\gamma^t$. (Our quantity $\mathbf{y}$ is the same as $\lambda^{t+1}$ and $\boldsymbol{\mu}^t$ is $\lambda^t$ as per their notation).

$$\gamma^t \leq \max\left\{\exp\left(\frac{-\|\boldsymbol{\mu}^t\|^2}{2\sigma^2}\right), \exp\left(\frac{-\langle\boldsymbol{\mu}^t, \boldsymbol{\mu}^\star\rangle^2}{2\|\boldsymbol{\mu}^\star\|^2\sigma^2}\right)\right\}. \tag{11}$$

Suppose $\mathcal{I}_t = 1$, then we have $\|\boldsymbol{\mu}^t - \boldsymbol{\mu}^\star\| \leq \frac{C\sigma}{10}$. That gives us that $\|\boldsymbol{\mu}^t\| \geq \frac{9}{10}C\sigma$ and $\langle\boldsymbol{\mu}^t, \boldsymbol{\mu}^\star\rangle \geq \frac{99}{50}C^2\sigma^2$. Together, this gives us that $\gamma^t \leq \frac{1}{8C^2}$. $\qquad\square$

Now that we have an expression for $\mathbf{y^t}$, we can plug it back into (10) to complete the proof.

$$\widehat{E}_{t+1} \leq (1-\eta)^2\tilde{E}_t + 4\lambda^t\eta(1-\eta)\tilde{E}_t$$
$$+ \eta^2\left(\sigma^2 d + 2\gamma^t\langle\boldsymbol{\mu}^\star - \boldsymbol{\mu}^t, \boldsymbol{\mu}^\star\rangle\right).$$

The inequality follows from the fact that $(2w^t(\mathbf{x}) - 1) \leq 1$ and $\mathbb{E}[\|\mathbf{x}\|^2] = \|\boldsymbol{\mu}^\star\|^2 + d\sigma^2$. Since $2\langle\boldsymbol{\mu}^\star - \boldsymbol{\mu}^t, \boldsymbol{\mu}^\star\rangle \leq \|\boldsymbol{\mu}^t - \boldsymbol{\mu}^\star\|^2 + \|\boldsymbol{\mu}^\star\|^2$, we get

$$\widehat{E}_{t+1} \leq \left((1-\eta)^2 + 4\lambda^t\eta + \lambda^t\eta^2\right)\tilde{E}_t + \eta^2\left(\sigma^2 d + 2\lambda^t\|\boldsymbol{\mu}^\star\|^2\right)$$
$$\leq \left(1 - \frac{3\eta}{8}\right)\tilde{E}_t + O(1)\eta^2\sigma^2 d,$$

where the last inequality follows from $\lambda^t \leq \frac{1}{8C^2}$ (Lemma 16). $\qquad\square$

We have now shown error reduction in single iteration at step $t + 1$, assuming that $\mathcal{I}_t$ holds.

In order to complete the proof of the main theorem, we require the following martingale lemma.

**Lemma 17.** *Suppose our initial estimates $\boldsymbol{\mu}^0$ satisfy $\|\boldsymbol{\mu}^0 - \boldsymbol{\mu}^\star\| \leq \frac{C\sigma}{20}$, then $\mathcal{I}_t = 1$ w.p $1 - O(\frac{1}{\text{poly}(N)})$, $\forall\, 1 \leq t \leq N$.*

*Proof.* This proof is identical to that of Theorem 8, where we use Azuma Hoeffding inequality to bound the sum of independent sub-Gaussian random variables. The only change is that the random variable $g^t(\mathbf{x})$ is replaced by $2w^t(\mathbf{x}) - 1$, while obtaining the sub-gaussian parameters. We omit the details from this presentation. $\qquad\square$

Using this martingale lemma, we can relate the quantity $\widehat{E}_{t+1}$ and $E_{t+1}$ similar to what we did for hard updates. Summing over $N$ steps, setting $\eta = \frac{3\log N}{N}$ and observing that maximum error when $\mathcal{I}_N = 0$ is $\|\boldsymbol{\mu}^\star\|^2$ gives the final result. $\qquad\square$