[Reviews · NeurIPS 2017]

Reviewer 1



I think that this is a good paper, on a timely subject: the estimation of centres in a Gaussian mixture model, the streaming setting, and the theoretical guarantees obtained on a non-convex optimisation problem are all very interesting. My two remarks would be the following: - In the discussion following theorem 1, is zeta (the exponent of N in the first additive term) a constant greater than 1? It is only claimed that it is 'large', but that is not exactly clear, and could change the impact of the result. - In any discussion regarding the lower bounds (intrinsic limitations of this problem), it is only explained that a constant fraction of the points will be attributed to the wrong cluster. It is not clear however that this will necessarily imply that the corresponding cluster is not well estimated. It would be interesting to comment on that. - p.2 'steaming' -> 'streaming'

Reviewer 2



The authors consider the problem of learning mixture of spherical Gaussians in the streaming setup. In particular, they assume a (usual) separation condition on the mixture components, and show that the mixture can be efficiently recovered in the streaming setup. Providing efficient algorithms for learning mixture of Gaussians is an active line of research. The authors extend the state-of-the-art for learning spherical GMMs by studying the streaming setting. The paper is well-written, and the claims seem to be sound. The provided analysis seems to show that the algorithm works when all the mixing weights are equal. The authors claim this can be extended to arbitrary mixing weights, yet I could not find sufficient support for this claim. The sample complexity of the method -- as the authors also suggest -- does not seem to be optimal. It would be nice to compare it with the best known bounds for the offline case [e.g., see https://arxiv.org/pdf/1706.01596.pdf].

Reviewer 3



This paper proposes a LLoyd type method with PCA initialization to estimate means of Gaussians in a streaming setting. References seem to be severely lacking as the domain is wide. The key point of the algorithm seems to be the initialization and there is no discussion on this part (comparison with the literature on k-means initialization). While the technical results might be interesting I have difficulties commenting on them without the proper background on the subject. I find as well that the technical exposition of proofs is not very clear (notations, chaining of arguments). Other concerns 1. Line 116 : What about the estimation of variance and weights? 2. Algorithm 1 : How do you know N in a streaming setting? 3. Theorem 1 : The error bound assumes that estimated centers ans true centers are matched. The bound is a k-means type bound. Hence, the title of the paper is misleading as it should talk about k-means clustering of Gaussian mixtures. The estimation of sigma is vaguely discussed in Section 6. 4. Section 6 looks like a very partial result with no appropriate discussion on why it is interesting.